# CATAPULTS IN SGD: SPIKES IN THE TRAINING LOSS AND THEIR IMPACT ON GENERALIZATION THROUGH FEATURE LEARNING

## ABSTRACT

In this paper, we first present an explanation regarding the common occurrence of spikes in the training loss when neural networks are trained with stochastic gradient descent (SGD). We provide evidence that the spikes in the training loss of SGD are "catapults", an optimization phenomenon originally observed in GD with large learning rates in Lewkowycz et al. (2020). We empirically show that these catapults occur in a low-dimensional subspace spanned by the top eigenvectors of the tangent kernel, for both GD and SGD. Second, we posit an explanation for how catapults lead to better generalization by demonstrating that catapults promote feature learning by increasing alignment with the Average Gradient Outer Product (AGOP) of the true predictor. Furthermore, we demonstrate that a smaller batch size in SGD induces a larger number of catapults, thereby improving AGOP alignment and test performance.

## 1 INTRODUCTION

Training algorithms are a key ingredient to the success of deep learning. Stochastic gradient descent (SGD) (Robbins & Monro, 1951), a stochastic variant of gradient descent (GD), has been effective in finding parameters that yield good test performance despite the complicated nonlinear nature of neural networks. Empirical evidence suggests that training networks using SGD with a larger learning rate results in better predictors (Frankle et al., 2019; Smith & Topin, 2019; Gilmer et al., 2021). In such settings, it is common to observe significant spikes in the training loss (Keskar & Socher, 2017; Ruder, 2016; Xing et al., 2018; LeCun et al., 2015) (see Figure 1 as an example). One may not *a priori* expect the training loss to decrease back to its "pre-spike" level after a large spike. Yet, this is what is commonly observed in training. Furthermore, the resulting "post-spike" model can yield improved generalization performance (He et al., 2016; Zagoruyko & Komodakis, 2016; Huang et al., 2017).

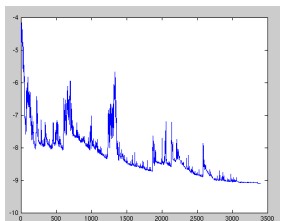

Figure 1: Spikes in training loss when optimized using SGD (x-axis: iteration). (Source: Wikipedia)

*Why do spikes occur during training, and how do the spikes affect generalization?*

In this work, we answer these question by connecting three common but seemingly unrelated phenomena in deep learning:

1. Spikes in the training loss of SGD,
2. Catapult dynamics in GD (Lewkowycz et al., 2020),
3. Better generalization when training networks with small batch SGD as opposed to larger batch size or GD.

In particular, we show that spikes in the training loss of SGD correspond to catapults, which were originally characterized in Lewkowycz et al. (2020) as a single spike in the loss when training with GD and large learning rate. We then show that smaller batch size in SGD results in a greater number of catapults. We connect the optimization phenomena of catapults to generalization by showing that catapults improve generalization through increasing feature learning, which is quantified by

the alignment between the Average Gradient Outer Product (AGOP) of the trained network and the true AGOP (Trivedi et al., 2014; Radhakrishnan et al., 2022; Xia et al., 2002; Härdle & Stoker, 1989; Hristache et al., 2001). Since decreasing batch size in SGD leads to more catapults, our result implies that SGD with small batch size yields improved generalization (see Table 1 for an example). We outline our specific contributions in the context of optimization and generalization below.

**Optimization.** We demonstrate that spikes in the training loss, specifically Mean Squared Error, occur in the top eigenspace of the Neural Tangent Kernel, a kernel resulting from the linearization of a neural network (Jacot et al., 2018). Namely, we project the residual (i.e., the difference between the predicted output and the target output) to the top eigenspace of the tangent kernel and show that spikes in the total loss function correspond to the spikes in the components of the loss in this low-dimensional subspace (see Section 3.1). In contrast, the components of the loss in the space spanned by the remaining eigendirections decrease monotonically. Thus, the catapult phenomenon occurs in the span of the top eigenvectors while the remaining eigendirections are not affected. This explains why the loss drops quickly to pre-spike levels, namely the loss value right before the spike, from the peak of the spike. We further show that multiple catapults can be generated in GD by increasing the learning rate during training (see Section 3.2). While prior work Lewkowycz et al. (2020) observed that the spectral norm of the tangent kernel decreased for one catapult, we extend that observation by showing that the norm decreases after each catapult.

We further provide evidence for catapults in SGD with large learning rates (see Section 3.3). Namely, we demonstrate that spikes in the loss when training with SGD correspond to catapults by showing that similarly to GD:

1. The spikes occur in the top eigenspace of the tangent kernel,
2. Each spike results in a decrease in the spectral norm of the tangent kernel.

We corroborate our findings across several network architectures including Wide ResNet (Zagoruyko & Komodakis, 2016) and ViT (Dosovitskiy et al., 2020) and datasets including CIFAR-10 (Krizhevsky et al., 2009) and SVHN (Netzer et al., 2011).

Moreover, as small batch size leads to higher variance in the eigenvalues of the tangent kernel for any given batch, small batch size results in an increased number of catapults.

**Generalization.** We show that catapults improve the generalization performance by alignment between the AGOP of the trained network with that of the true model[1]. The AGOP identifies the features that lead to greatest change in predictor output when perturbed and has been recently identified as the mechanism through which neural networks learn features (Radhakrishnan et al., 2022). We use AGOP alignment to provide an explanation for prior empirical results from Zhu et al. (2022b); Lewkowycz et al. (2020) showing that a single catapult can lead to better test performance in GD. Moreover, we extend these prior results to show that test performance continues to improve

| Batch size | AGOP alignment | Test loss |
|---|---|---|
| 2000 (GD) | 0.81 | 0.74 |
| 50 | 0.84 | 0.71 |
| 10 | 0.89 | 0.59 |
| 5 | 0.95 | 0.42 |

Table 1: Smaller SGD batch size leads to a higher (better) AGOP alignment and smaller (better) test loss. The results correspond to Figure 7a (a synthetic dataset).

as the number of catapults increases in GD. Thus, we show that decreasing batch size with SGD can lead to better test performance due to an increase in the number of catapults. We further demonstrate that AGOP alignment is an effective measure of generalization by showing that test error is highly correlated with the AGOP alignment when training on the same task across different optimization algorithms including Adagrad (Duchi et al., 2011), Adadelta (Zeiler, 2012) and Adam (Kingma & Ba, 2014) etc. We corroborate our findings on CelebA (Liu et al., 2015) and SVHN (Netzer et al., 2011) datasets and architectures including fully-connected and convolutional neural networks. See Section 4.

## 1.1 RELATED WORKS

**Linear dynamics.** Recent studies have shown that (stochastic) GD for wide neural networks provably converges to global minima with an appropriately small learning rate (Du et al., 2019; Liu et al., 2022; Zou & Gu, 2019). These works leveraged the fact that neural networks, under certain conditions on initialization, can be accurately approximated by their linearization when network width is

---

[1]When the true model is not available, we use a SOTA model as a substitute.

sufficiently large (Jacot et al., 2018; Liu et al., 2020; Zhu et al., 2022a; Liu et al., 2021). In this setting, referred to as the kernel regime, the training dynamics of wide networks can be approximated by the linear dynamics of the corresponding linear model.

**Catapult phase.** When training networks with GD and large learning rate, recent work Lewkowycz et al. (2020) identified a striking phenomenon that cannot be manifested in the kernel regime. This phenomenon, referred to as the "catapult phase", is characterized by an increase in loss during the early stages of training, followed by a decrease that forms a single spike in the training loss. After the catapult, the spectral norm of the tangent kernel, i.e., its top eigenvalue, decreases and thus, remarkably prevents divergence. Recent studies focusing on understanding catapults in GD include Zhu et al. (2022b), which considers quadratic approximations of neural networks, and Meltzer & Liu (2023), examining two-layer homogeneous neural networks. Our work investigates the impact of catapults in SGD on both optimization and generalization through experimental approaches.

**Edge of stability.** A phenomenon related to catapults is the "Edge of Stability" (EoS), which describes the dynamics of the training loss and the sharpness, i.e., eigenvalues of the Hessian of the loss, at the later stage of training networks with GD (Cohen et al., 2020) and SGD (Jastrzebski et al., 2019; Jastrzębski et al., 2018). In Cohen et al. (2020), it was conjectured that at EoS, the spikes in the training loss are micro-catapults. There is a growing body of work analyzing the mechanism of EoS in training dynamics with GD (Arora et al., 2022; Ahn et al., 2022; Damian et al., 2022; Li et al., 2022; Agarwala et al., 2022; Agarwala & Dauphin, 2023), and SGD Kalra & Barkeshli (2023). Our work provides evidence that the spikes in the training loss using SGD are catapults and demonstrates the connection between the loss spikes and feature learning.

**Generalization and sharpness.** It has been observed that networks trained with SGD generalize better than GD, and smaller batch sizes often lead to better generalization performance (Kandel & Castelli, 2020; LeCun et al., 2002; Masters & Luschi, 2018; Keskar et al., 2016; Smith et al., 2020; Goyal et al., 2017; Jastrzębski et al., 2017). Empirically, it has been observed that training with SGD results in flat minima (Hochreiter & Schmidhuber, 1994; 1997). However, we noticed that it is not always the case, e.g., Geiping et al. (2021). A number of works been argued that flatness of the minima is connected to the generalization performance (Neyshabur et al., 2017; Wu et al., 2017; Kleinberg et al., 2018; Xie et al., 2020; Jiang et al., 2019; Dinh et al., 2017), however we know only one theoretical result in that direction (Ding et al., 2022). Training algorithms aiming to find a flat minimum were developed and shown to perform well on a variety of tasks (Foret et al., 2020; Izmailov et al., 2018). As an explanation for empirically observed improved generalization, prior work Lewkowycz et al. (2020) argued that a single catapult with GD resulted in flatter minima. In this work we propose a different line of investigation to understand generalization properties of GD-based algorithms based on feature learning as measured by the alignment with AGOP.

## 2 PRELIMINARIES

**Notation.** We use bold letters (e.g.,$\mathbf{w}$) to denote vectors and capital letters (e.g.,$K$) to denote matrices. For trainable parameters, we use superscript $t$ to denote the iteration $t$, e.g., $\mathbf{w}^t$ during training. We use $\tilde{O}(\cdot)$ to denote the same order of magnitude as $O(\cdot)$ but with logarithmic factors absorbed. We use $\|\cdot\|_F$ to denote the Frobenius norm and use $\|\cdot\|_2$ to denote the spectral norm.

**Optimization task.** Given training data $\{(\boldsymbol{x}_i, y_i)\}_{i=1}^n := (X, \mathbf{y})$ with data $\boldsymbol{x}_i \in \mathbb{R}^d$ and labels $y_i \in \mathbb{R}$ for $i \in [n]$, we minimize the Mean Square Error (MSE) $\mathcal{L}(\mathbf{w}; (X, \mathbf{y})) = \frac{1}{n}\sum_{i=1}^n (f(\mathbf{w}; \boldsymbol{x}_i) - y_i)^2$, where $f(\mathbf{w}; \cdot) : \mathbb{R}^p \to \mathbb{R}$ is a parameterized model, e.g., a neural network. We denote the weight parameters at initialization by $\mathbf{w}_0$. Mini-batch SGD is conducted as follows:

$$\mathbf{w}^{t+1} = \mathbf{w}^t - \frac{\eta}{b}\frac{\partial}{\partial \mathbf{w}}\sum_{\boldsymbol{x}_j \in X_{\text{batch}}} (f(\mathbf{w}^t; \boldsymbol{x}_j) - y_j)^2,$$

where $\eta$ is the learning rate and $(X_{\text{batch}}, \mathbf{y}_{\text{batch}})$ is a randomly sampled batch from $(X, \mathbf{y})$ with batch size $b$. When $b = n$, mini-batch SGD reduces to GD.

**Paramterization.** Unless specified, we use NTK parameterization (Jacot et al., 2018) for neural networks, that is, we initialize each trainable parameter i.i.d. from standard normal distribution, i.e., $\mathbf{w}_0 \sim \mathcal{N}(\mathbf{0}, I_p)$, and multiply each pre-activated neuron with an extra scaling factor $1/\sqrt{m}$ where $m$ is the fan-in of that neuron. For example, for neurons in fully connected networks, the fan-in is the width of the previous hidden layer. In convolutional nets, fan-in is the number of channels multiplied by the size of the window. We also use Pytorch (Paszke et al., 2019) default parameterization in some settings.

**Definition 1** ((Neural) Tangent Kernel). *The tangent kernel $K(\mathbf{w}; \cdot, \cdot)$ of a parameterized machine learning model $f(\mathbf{w}; \cdot) : \mathbb{R}^p \times \mathbb{R}^d \to \mathbb{R}$ is defined as*

$$K(\mathbf{w}; \boldsymbol{x}, \boldsymbol{z}) = \left\langle \partial f(\mathbf{w}; \boldsymbol{x}) \big/ \partial \mathbf{w}, \partial f(\mathbf{w}; \boldsymbol{z}) \big/ \partial \mathbf{w} \right\rangle, \quad \forall \boldsymbol{x}, \boldsymbol{z} \in \mathbb{R}^d. \tag{1}$$

**Critical learning rates.** For a wide neural network $f(\mathbf{w}; \cdot)$ with input data $X$ trained with GD, the optimization dynamics will follow catapult dynamics when the learning rate is larger than a critical learning rate (Lewkowycz et al., 2020). Otherwise, dynamics approximately follow linear dynamics where the loss decreases monotonically (Lee et al., 2019). The critical learning rate is defined as

$$\eta_{\mathrm{crit}}(f(\mathbf{w}_0; X)) := 2 \big/ \lambda_{\max}(H_{\mathcal{L}}(\mathbf{w}_0)), \tag{2}$$

where $H_{\mathcal{L}}$ denotes the Hessian of the loss.

Note that for a linear model $h(\mathbf{w}; \cdot) = \langle \mathbf{w}; \phi(\cdot) \rangle : \mathbb{R}^p \times \mathbb{R}^d \to \mathbb{R}$ with input data $X \in \mathbb{R}^{n \times d}$, the tangent kernel is given by $K = \phi(X)\phi(X)^T$ and the Hessian is $H_{\mathcal{L}} = \frac{2}{n}\phi(X)^T\phi(X)$. Therefore, $K$ and $H$ share the same largest eigenvalue, and we have $\eta_{\mathrm{crit}}(h) = n/\lambda_{\max}(K)$. As wide networks are close to their linear approximations at initialization, we can further approximate $\eta_{\mathrm{crit}}$ by $\tilde{\eta}_{\mathrm{crit}} = n/\lambda_{\max}(K(\mathbf{w}_0))$ where $|\eta_{\mathrm{crit}} - \tilde{\eta}_{\mathrm{crit}}| = \tilde{O}(1/\sqrt{m})$ with $m$ denoting network width (see the derivation in Appendix A.1). In principle, the critical learning rate can be defined pointwise for any $\mathbf{w}$ for which the network $f(\mathbf{w}; \cdot)$ is close to its linear approximation. When it is clear from the context, we omit the argument of $\eta_{\mathrm{crit}}$. Lastly, we use $\eta_{\max}$ to denote the maximum learning rate with which the optimization of the loss function can stably converge (possibly through catapults).

## 3 CATAPULTS IN OPTIMIZATION

### 3.1 CATAPULTS OCCUR IN THE TOP EIGENSPACE OF THE TANGENT KERNEL FOR GD

In this section, we show that catapults mainly occur in the subspace spanned by the top eigen-directions of the tangent kernel.

Given input data $X \in \mathbb{R}^{n \times d}$, let $\mathbf{f}^t \in \mathbb{R}^n$ denote the predictions of $f$ on $X$ at iteration $t$. The tangent kernel matrix $K(\mathbf{w}^t) \in \mathbb{R}^{n \times n}$ corresponding to $\mathbf{f}^t$ can be decomposed as $K(\mathbf{w}^t) := K^t = \sum_{j=1}^n \lambda_j^t \mathbf{u}_j^t \mathbf{u}_j^{t\,T}$, with $\lambda_j^t$ and $\mathbf{u}_j^t \in \mathbb{R}^n$, $j \in \{1, \cdots, n\}$, being the eigenvalues and unit-length eigenvectors, respectively. We assume $\lambda_1 \geq \lambda_2 \geq \cdots \geq \lambda_n$. Given an integer $s$, $1 \leq s < n$, we consider the *top eigenspace*, i.e., the subspace spanned by the top eigenvalues $\lambda_j$ with $1 \leq j \leq s$, as well as the corresponding projection operators $\mathcal{P}_s : \mathbb{R}^n \to \mathbb{R}^n$.

For the residual vector $\mathbf{r}^t := \mathbf{f}^t - \mathbf{y}$, define $\mathcal{P}_s \mathbf{r}^t = \sum_{j=1}^s \left\langle \mathbf{r}^t, \mathbf{u}_j^t \right\rangle \mathbf{u}_j^t$ as the projection of $\mathbf{r}^t$ onto the top eigenspace and $\mathcal{P}_s^\perp \mathbf{r}^t = \sum_{j=s+1}^n \left\langle \mathbf{r}^t, \mathbf{u}_j^t \right\rangle \mathbf{u}_j^t$ as the projection onto the complementary subspace. We can decompose the loss $\mathcal{L}(\mathbf{f}^t)$ as

$$\mathcal{L}(\mathbf{f}^t) = \frac{1}{n} \big\| \mathbf{f}^t - \mathbf{y} \big\|_2^2 = \frac{1}{n} \big\| \mathcal{P}_s \mathbf{r}^t \big\|_2^2 + \frac{1}{n} \big\| \mathcal{P}_s^\perp \mathbf{r}^t \big\|_2^2, \tag{3}$$

We denote $\frac{1}{n} \big\| \mathcal{P}_s \mathbf{r}^t \big\|_2^2$ and $\frac{1}{n} \big\| \mathcal{P}_s^\perp \mathbf{r}^t \big\|_2^2$ by $\mathcal{PL}_s$ and $\mathcal{PL}_s^\perp$ respectively and hide $t$ for simplicity.

We now present our claim that the catapult occurs in the top eigenspace of the tangent kernel.

**Claim 1.** *For $\eta \in (\eta_{\mathrm{crit}}, \eta_{\max})$, the catapult occurs in the top eigenspace of the tangent kernel, i.e., the loss spike in $\mathcal{PL}_s^\perp$ diminishes in magnitude as $s$ increases.*

We empirically justify Claim 1. In particular, we consider two neural network architectures: a 5-layer Fully Connected Neural Network (FCN) and a 5-layer Convolutional Neural Network (CNN). We train the models with fixed learning rates on CIFAR-2 (2 class subset of CIFAR-10) and take $n = 128$ and $s = 1, 3, 5$. The experimental details can be found in Appendix F. From the results in Figure 2, we can see that when there is a catapult, as $s$ increases, the loss value at the peak of the spike in $\mathcal{PL}_s^\perp$ decreases. Finally, $\mathcal{PL}_5$ corresponds to the spike in the training loss while $\mathcal{PL}_5^\perp$ decreases almost monotonically.

**Choice of top eigenspace dimension $s$.** Note that a larger learning rate will make the loss increase in more eigen-directions, as indicated by the linear dynamics before the spike. Therefore, in general, we need a larger $s$ to capture the spikes in the training loss for larger learning rates. However, for $\eta \in (\eta_{\mathrm{crit}}, \eta_{\max})$ such that the GD can stably converge with catapults, we consistently observe that $s$ is a small constant no larger than 10 in all our experiments. The remaining loss $\mathcal{PL}_s^\perp$ decreases nearly monotonically with the number of iterations (See Figure 2b and d).

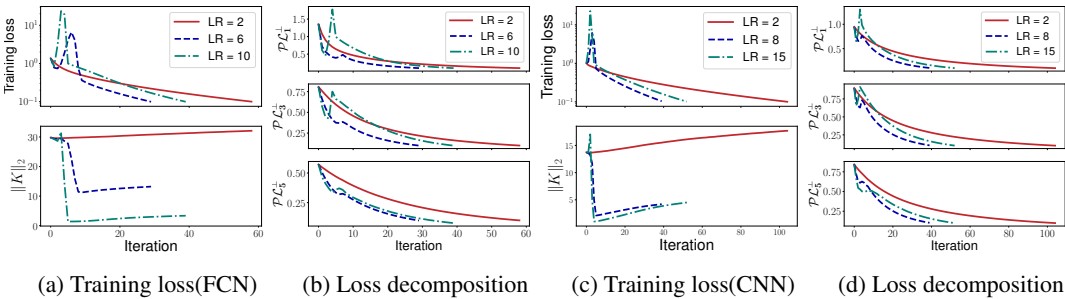

(a) Training loss(FCN)    (b) Loss decomposition    (c) Training loss(CNN)    (d) Loss decomposition

Figure 2: **Catapult dynamics for 5-layer FCN (a-b) and CNN (c-d)**. Panel (a) and (c) are training loss and the spectral norm of the tangent kernel with different learning rates, and Panel (b) and (d) are training loss decomposed into non-top eigenspace of the tangent kernel, $\mathcal{PL}_1^\perp, \mathcal{PL}_3^\perp$ and $\mathcal{PL}_5^\perp$. All the networks are trained on a subset of CIFAR-10. In this experiment, the critical learning rates for FCN and CNN are $3.6$ and $4.5$ respectively.

Lastly, we note that our claim extends to multidimensional outputs. In particular, for $k$-class classification tasks, we project the flattened vector of predictions of size $kn$ to the top eigenspaces of the empirical NTK, which is of size $kn \times kn$. Correspondingly, we empirically observe that catapults occurs in the top $ks$ eigenspace with a small $s$. See Figure 10 in Appendix B.

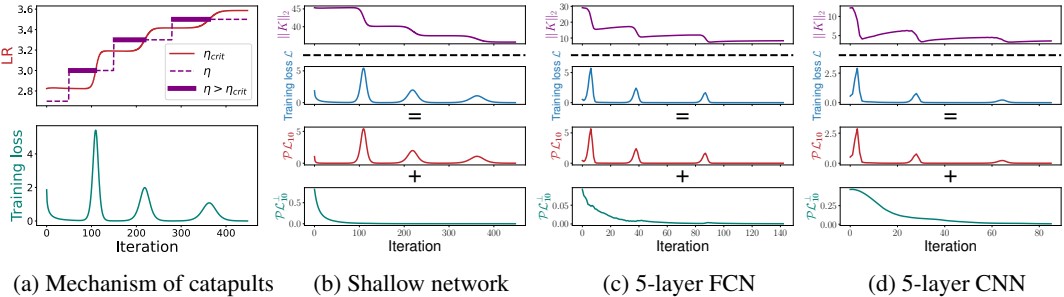

(a) Mechanism of catapults    (b) Shallow network    (c) 5-layer FCN    (d) 5-layer CNN

Figure 3: **Multiple catapults during GD with increased learning rates**. Panel (a): the mechanism of multiple catapults: extra spike in the training loss occurs when $\eta$ is increased to satisfy $\eta > \eta_{\text{crit}}$. Panel (b,c,d): decomposing the training loss of the shallow network, 5-layer FCN, and 5-layer CNN respectively into $\mathcal{PL}_{10}$ and $\mathcal{PL}_{10}^\perp$. The experimental details can be found in Appendix F.2.

### 3.2 MULTIPLE CATAPULTS IN GD

Much empirical evidence has shown that during the catapult dynamics, there is a single spike in the training loss (Lewkowycz et al., 2020). In this section, we show that during GD, the spike in the training loss can be generated multiple times by repeatedly increasing the learning rate during training. As a result, $\|K\|_2$ will accordingly decrease multiple times.

In the experiments, we start with a small constant learning rate, i.e., $\eta < \eta_{\text{crit}} \approx n/\lambda_{\max}(K)$. We increase the learning rate to let $\eta > \eta_{\text{crit}}$ to generate the catapult. We repeat this procedure multiple times to generate multiple catapults. Each time we use a larger $\eta$, as $\eta_{\text{crit}}$ becomes larger after each catapult. This mechanism is illustrated in Figure 3(a) in which we train a wide, shallow network on 128 data points from CIFAR-2. In Figure 3b, we can see that for the shallow network, $\mathcal{PL}_{10}$ captures the spikes of the loss.

We further use the same 5-layer FCN and 5-layer CNN as the ones in Figure 2 and show that multiple catapults can be seen in deep nets as well since (1) the spectral norm of the tangent kernel decreases corresponding to each spike and (2) all the spikes in the training loss are mainly in the top eigenspace of the tangent kernel (See Figure 3b and c).

### 3.3 CATAPULTS IN SGD

In this section, we provide evidence that catapults also occur in SGD and manifest as spikes in the training loss. Specifically, we leverage the mechanism of catapults in GD to demonstrate the

occurrence of catapults in batches of SGD, and we further show that the empirical phenomenon observed in the catapults of GD occurs in SGD.

**Mechanism of catapults in SGD.**   Our analysis of multiple catapults in GD made clear that the mechanism of the catapult lies in the quantitative relation between $\eta$ and $\eta_{\text{crit}}$. Namely, the catapult occurs when $\eta > \eta_{\text{crit}}$ otherwise the loss decreases monotonically. In SGD, for a fixed large learning rate $\eta$, the critical learning rate $\eta_{\text{crit}}$ defined on each batch will oscillate around $\eta$ due to the variance of batches. Therefore, we can expect that there are numerous catapults in SGD, which ultimately form spikes in the training loss.

We mathematically formulate this reasoning as follows. For a batch $X_{\text{batch}} \subset X$, we can similarly find the critical learning rate for each batch $\eta_{\text{crit}}(X_{\text{batch}}) := 2/\lambda_{\max}(H_{\mathcal{L}}(\mathbf{w}; X_{\text{batch}}))$. We investigate the change of the loss after one SGD step corresponding to this batch since an SGD step can be viewed as a GD step on a batch. We further consider the loss change corresponding to the top eigenspace of the tangent kernel, since that is where the catapults occur (as shown in Section 3.1):

$$\mathcal{P}\text{Diff}_1(\mathbf{f}^t(X_{\text{batch}})) := \mathcal{PL}_1(\mathbf{f}^{t+1}(X_{\text{batch}})) - \mathcal{PL}_1(\mathbf{f}^t(X_{\text{batch}})). \tag{4}$$

Following the same reasoning for GD, we make the following claim for SGD:

**Claim 2.** *For neural networks* $\mathbf{f}$ *with width* $m$, $\forall t > 0$, $\text{sgn}(\eta - \eta_{\text{crit}}(\mathbf{f}^t(X_{\text{batch}})) + \varepsilon) = \text{sgn}(\mathcal{P}\text{Diff}_1(\mathbf{f}^t(X_{\text{batch}})))$ *where* $|\varepsilon| = \tilde{O}(1/\sqrt{m})$.

**Remark 1.** *For linear model* $\mathbf{f}(\cdot)$, *the claim holds with* $\varepsilon = 0$. *For wide neural networks,* $\varepsilon$ *measures the deviation from their pointwise linear approximations (Lee et al., 2019; Liu et al., 2020).*

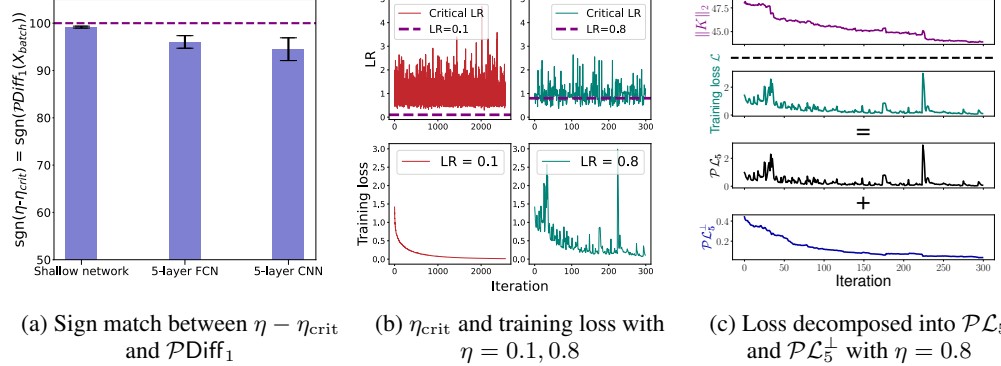

(a) Sign match between $\eta - \eta_{\text{crit}}$ and $\mathcal{P}\text{Diff}_1$

(b) $\eta_{\text{crit}}$ and training loss with $\eta = 0.1, 0.8$

(c) Loss decomposed into $\mathcal{PL}_5$ and $\mathcal{PL}_5^{\perp}$ with $\eta = 0.8$

Figure 4: **Mechanism of catapults in SGD.** Panel(a): The alignment between $\text{sgn}(\eta - \eta_{\text{crit}})$ and $\text{sgn}(\mathcal{P}\text{Diff}_1^t)$, for which the catapults occur (see the training loss in Figure 4(c) for shallow net and Figure 5 (a,b) for deep nets). Panel(b): $\eta_{\text{crit}}(X_{\text{batch}})$ and training loss when trained with $\eta = 0.1, 0.8$. Panel (c): training loss with $\eta = 0.8$ decomposed into $\mathcal{PL}_5$ and $\mathcal{PL}_5^{\perp}$. All the networks are trained on a subset of CIFAR-10. See the details of experiments in Appendix F.3.

We empirically verify our claim on the following finite width networks: a shallow network and the same 5-layer FCN and CNN as the ones in Figure 2. To ensure the occurrence of catapults, we choose the learning rate as the critical learning rate of the whole training set. In Figure 4a, we show the probability of $\text{sgn}(\mathcal{P}\text{Diff}_1) = \text{sgn}(\eta - \eta_{\text{crit}})$ throughout the training. Our results demonstrate that the sign of $\mathcal{P}\text{Diff}_1(\mathbf{f}^t(X_{\text{batch}}))$ and $\eta - \eta_{\text{crit}}(\mathbf{f}^t(X_{\text{batch}}))$ is well matched throughout the training process thus corroborating the existence of catapult dynamics in SGD. As $\tilde{\eta}_{\text{crit}} = b/\|K(\mathbf{w}; X_{\text{batch}})\|_2$ is close to $\eta_{\text{crit}}$ for wide neural networks (see the empirical validation in Appendix A), where $b$ denotes the batch size, similar to GD, we can use $\|K(\mathbf{w}; X_{\text{batch}})\|_2$ to indicate the change of $\eta_{\text{crit}}$ for SGD.

We now consider the spikes in the training loss of SGD. We can expect that the training loss is likely to increase and result in a spike when $\mathcal{PL}_1(X_{\text{batch}})$ increases since $\nabla_{\mathbf{w}}\mathcal{L}(X_{\text{batch}})$ approximates $\nabla_{\mathbf{w}}\mathcal{L}(X)$. As $\mathcal{PL}_1(X_{\text{batch}})$ increases when $\eta > \eta_{\text{crit}}$ for the selected batch, we analyze the relationship between when $\eta > \eta_{\text{crit}}$ across batches and the emergence of spikes. Specifically, we train a shallow network with a learning rate of 0.1, which is always below $\eta_{\text{crit}}(X_{\text{batch}})$ (Figure 4b upper left), and a learning rate of 0.8, which oscillates around $\eta_{\text{crit}}(X_{\text{batch}})$ (Figure 4b upper right). We observe that the training loss has significant spikes with $\eta = 0.8$, but decreases monotonically with

$\eta = 0.1$ (Figure 4b). This suggests that spikes in the training loss are catapults. Below, we provide additional evidence that such spikes in the training loss of SGD are indeed catapults.

**Catapults occur in the top eigenspace of the tangent kernel for SGD.** We have shown in Section 3.1 that in GD the catapults consistently occur in the top eigenspace of the tangent kernel. Now we show that we observe a similar pattern in SGD.

In the experiment for shallow networks, we similarly decompose the loss with $\eta = 0.8$ into $\mathcal{PL}_5$ and $\mathcal{PL}_5^\perp$ based on the eigenspace of the tangent kernel. We observe that $\mathcal{PL}_5$ corresponds to the spikes in the training loss, while the decrease of $\mathcal{PL}_5^\perp$ is nearly monotonic, with only small oscillations present (Figure 4c).

We also observe the same phenomenon for deep networks. Specifically, $\mathcal{PL}_1$ corresponds to the spikes in the training loss while the remaining loss $\mathcal{PL}_1^\perp$ decreases nearly monotonically. The empirical results are shown in Figure 5, where we consider four network architectures: (1) 5-layer FCN, (2) 5-layer CNN (the same as the ones in Figure 2), (3) Wide ResNets 10-10 (Zagoruyko & Komodakis, 2016) and (4) ViT-4 (Dosovitskiy et al., 2020).

This observation, along with the results in Figure 5 for deep networks, is consistent with our findings for GD and provides evidence that the spikes in training loss for neural networks are caused by catapults.

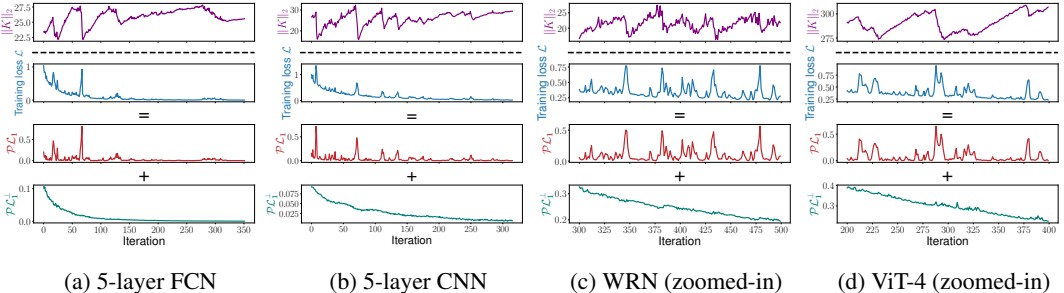

(a) 5-layer FCN      (b) 5-layer CNN      (c) WRN (zoomed-in)      (d) ViT-4 (zoomed-in)

Figure 5: **Catapult dynamics in SGD for modern deep architectures.** The training loss is decomposed into $\mathcal{PL}_1$ and $\mathcal{PL}_1^\perp$. We train the networks on a subset of CIFAR-10. The complete versions of Panel (c) and (d) can be found in Figure 11 in Appendix B.

**Decreases in the spectral norm of the tangent kernel correspond to spikes.** In this section, we provide further evidence of catapults in SGD by empirically showing that the spectral norm of the tangent kernel decreases whenever there is a spike in SGD.

In Lewkowycz et al. (2020), the catapult led to a decrease in the spectral norm of the tangent kernel in GD, and our experimental results in Section 3.1 extended this finding to settings with multiple catapults in GD. We observe a consistent phenomenon in SGD. Specifically, in our experiments with shallow networks, we observe a significant decrease in the spectral norm of the tangent kernel whenever there is a spike in the training loss during SGD (Figure 4c). This observation remains consistent for deep networks (Figure 5) as well. Therefore, this finding further justifies that spikes in the training loss of SGD are catapults.

We further validate our empirical observations on (1) the occurrence of the loss spikes of SGD in the top eigenspace of the tangent kernel and (2) the decrease in the spectral norm of the tangent kernel during loss spikes in the setting with Pytorch default parameterization, under which the wide networks are still close to their linear approximations (Liu et al., 2020; Yang & Hu, 2020), and on additional datasets (see Appendix B). All experimental details can be found in Appendix F.

## 4    CATAPULTS LEAD TO BETTER GENERALIZATION THROUGH FEATURE LEARNING

Previous empirical results from Zhu et al. (2022b); Lewkowycz et al. (2020) show that a single catapult can lead to better test performance in GD for wide neural networks. In this section, we observe a similar trend in our experiments for both GD and SGD with multiple catapults. We further show that the underlying mechanism behind the improved test performance is the alignment between the trained network's AGOP and the true AGOP, which increases when catapults occur.

The AGOP is defined as follows:

**Definition 2** (Average Gradient Outer Product (AGOP)). *Given a parameterized model $f(\mathbf{w}; \cdot) : \mathbb{R}^p \times \mathbb{R}^d \to \mathbb{R}$, the AGOP of it with respect to data $X \in \mathbb{R}^{n \times d}$ is defined as*

$$G(\mathbf{w}) = \frac{1}{n} \sum_{i=1}^{n} \frac{\partial f(\mathbf{w}; \boldsymbol{x}_i)}{\partial \boldsymbol{x}_i} \frac{\partial f(\mathbf{w}; \boldsymbol{x}_i)}{\partial \boldsymbol{x}_i}^T \in \mathbb{R}^{d \times d}. \tag{5}$$

**Remark 2.** *The derivative of the predictor with respect to the input provides a natural measure of the significance of each element in the input to the predictor. Typically, important features will have a larger scale of derivatives since they lead to greater change in predictor output upon perturbation. The AGOP was recently identified as the mechanism through which neural networks select features and is thus directly a measure of feature learning in neural networks (Radhakrishnan et al., 2022).*

To measure feature learning for the trained model $f$, we quantify the degree of *AGOP alignment*. Specifically, we evaluate the cosine similarity between $G$ and optimal $G^*$ which corresponds to $f$ and $f^*$ respectively on the test set:

**AGOP alignment :**
$$\cos(G, G^*) := \frac{\langle G, G^* \rangle}{\|G\|_F \|G^*\|_F}, \tag{6}$$

where $f^*$ is the true model. When the true model is not available, we use a state-of-the-art model as a substitute.

**Experimental settings.** We work with a total of five datasets: three synthetic datasets and two real-world datasets. For synthetic datasets, we consider true functions $f^*(\boldsymbol{x}) = (1)x_1 x_2$ (rank-2), $(2)x_1 x_2 (\sum_{i=1}^{10} x_i)$ (rank-3) and (3) $\sum_{j=1}^{4} \prod_{i=1}^{j} x_i$ (rank-4) (Abbe et al., 2021). The true functions $f^*(\boldsymbol{x})$ of these datasets are multi-index models e.g., functions of the form $f^*(\boldsymbol{x}) = g(U\boldsymbol{x})$ where $U$ is a low-rank matrix. The relationship between AGOP and generalization on learning such functions was discussed in (Härdle & Stoker, 1989; Hristache et al., 2001; Trivedi et al., 2014; Xia et al., 2002; Radhakrishnan et al., 2022; Damian et al., 2022). For the two real-world datasets, we use (1) CelebA (Liu et al., 2015) and (2) SVHN dataset (Netzer et al., 2011). The experimental details can be found in Appendix F.

**Improved test performance by catapults in GD.** In Section 3.2, we showed that catapults can be generated multiple times. We now show that generating multiple catapults leads to improved test performance of neural networks trained with GD by leading to increased AGOP alignment. In Figure 6, we can see for all tasks, the test loss/error decreases as the number of catapults increases while AGOP alignment increases. This indicates that learning AGOP strongly correlates with test performance.

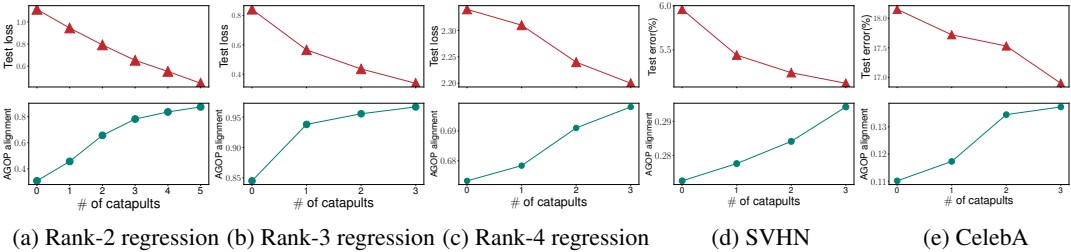

(a) Rank-2 regression (b) Rank-3 regression (c) Rank-4 regression     (d) SVHN     (e) CelebA

Figure 6: **Multiple catapults in GD.** A greater number of catapults in GD leads to a higher (better) AGOP alignment and smaller (better) test loss/error. We train 2-layer FCN in Panel(a), 4-layer FCN in Panel(b,c,e) and 5-layer CNN in Panel(d). Experimental details can be found in Appendix F.4.

**Intuition of the connection between catapults and AGOP alignment.** In particular, catapults occur when training with a large learning rate ($\eta \in (\eta_{\text{crit}}, \eta_{\text{max}})$), where the training dynamics are no longer in the NTK regime. Now a large learning rate diminishes the effect of initialization on the final weights. This is because the weight change scales with the loss, which is large when the catapults occur. Consequently, it leads to a more significant change in the gradient $df/d\boldsymbol{x}$. This is analogous to escaping the NTK regime and feature learning by utilizing near zero initialization as discussed in Yang & Hu (2020); Radhakrishnan et al. (2022), in which case the effect of initialization is negligible due to its small magnitude. We validate in experiments that using near-zero initialization leads to increased AGOP alignment and better generalization(See Figure **??** in Appendix). Thus, in summary, networks can be trained with such large learning rates due to the catapult phenomena and as an additional consequence, such large learning rates lead to improved AGOP alignment.

**Improved test performance by catapults in SGD.** In Section 3.3, we have demonstrated the occurrence of catapults in SGD. We now show that decreasing batch size in SGD leads to better test performance as a result of an increase in the number of catapults and thus, increased AGOP alignment. We estimate the number of catapults during training by counting the number of the occurrence of the event $\mathcal{P}\mathrm{Diff}_1^t(X_{\mathrm{batch}}) > \epsilon$ with $\epsilon = 10^{-8}$ until the best validation loss/error. Recall that when catapults occur, the component of the loss in the top eigenspace of the tangent kernel will increase, as discussed in section 3.3.

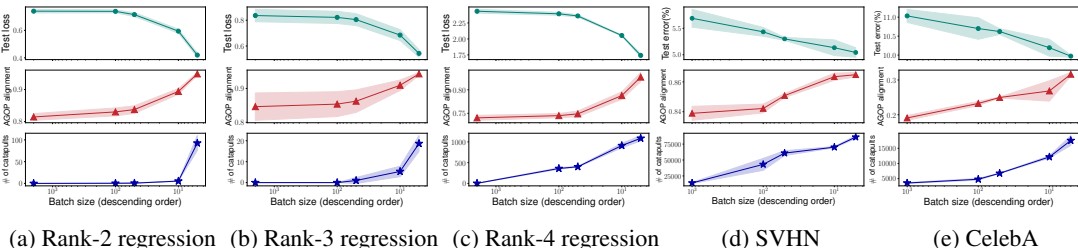

(a) Rank-2 regression  (b) Rank-3 regression  (c) Rank-4 regression     (d) SVHN          (e) CelebA

Figure 7: **Similarity between AGOP alignment and test performance.** A greater number of catapults in SGD leads to a higher (better) AGOP alignment and smaller (better) test loss/error. We train a 2-layer FCN in Panel(a), 4-layer FCN in Panel(b,c,e) and 5-layer CNN in Panel(d) by SGD.

In Figure 7, we can see that across all tasks, as the batch size decreases, (1) the number of catapults increases, (2) the test loss/error decreases and (3) the AGOP alignment improves. These findings indicate that in SGD, a smaller batch size leads to more catapults which in turn improves the test performance through alignment with the AGOP . These observations are consistent with our findings in GD. We further verify our observation with Pytorch default parameterization on the same tasks (see Figure 19 in Appendix E).

**Generalization with different optimizers correlates with AGOP alignment.** We further demonstrate the strong correlation between the test performance and AGOP alignment by comparing the predictors trained on the same task with a number of different optimization algorithms. From the results shown in Figure 8, we can see that the AGOP alignment strongly correlates with the test performance, which suggests that models learning the AGOP is useful for learning the problem.

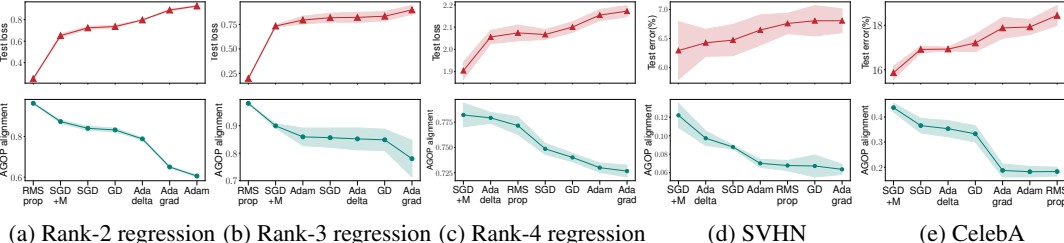

(a) Rank-2 regression (b) Rank-3 regression (c) Rank-4 regression     (d) SVHN          (e) CelebA

Figure 8: **Strong correlation between test performance and AGOP alignment for different optimization algorithms.** We train a 2-layer FCN in Panel(a), a 4-layer FCN in Panel(b,c,e) and a 5-layer CNN in Panel(d). We use GD, SGD, SGD with Momentum (Qian, 1999)(SGD+M), RMSprop (Hinton), Adagrad (Duchi et al., 2011), Adadelta (Zeiler, 2012) and Adam (Kingma & Ba, 2014) for training.

## 5 SUMMARY

In this work, we framed and answered two questions: (1) why do spikes occur during training with SGD and (2) how do the spikes affect generalization? For the first question, we demonstrate that the loss spikes correspond to catapults by showing the spikes occur in the top eigenspace of the tangent kernel and each loss spike corresponds to a decrease in the spectral norm of the tangent kernel. For the second question, we show that catapults lead to increased alignment between the AGOP of the trained model and the true AGOP. A consequence of our results is the explanation for the observation that SGD with small batch size leads to improved generalization. This is due to an increase in the number of catapults and thus improved AGOP alignment. For the future direction, it would be interesting to connect AGOP alignment with functional properties observed in neural networks, e.g., Jacot (2023).

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
