## APPENDIX

## A $\quad \eta_{\text{crit}} \approx \tilde{\eta}_{\text{crit}}$ FOR WIDE NEURAL NETWORKS

### A.1 DERIVATION FOR INITIALIZATION

For MSE $\mathcal{L}(\mathbf{w}; X) = \frac{1}{n} \sum_{i=1}^{n} (f(\mathbf{w}; \mathbf{x}_i) - y_i)^2$, we can compute its $H_{\mathcal{L}}$ by the chain rule:

$$H_{\mathcal{L}}(\mathbf{w}) = \underbrace{\frac{2}{n} \sum_{i=1}^{n} \left( \frac{\partial f(\mathbf{w}; \mathbf{x}_i)}{\partial \mathbf{w}} \right)^T \frac{\partial f(\mathbf{w}; \mathbf{x}_i)}{\partial \mathbf{w}}}_{\mathcal{A}(\mathbf{w})} + \underbrace{\frac{2}{n} \sum_{i=1}^{n} (f(\mathbf{w}; \mathbf{x}_i) - y_i) \frac{\partial^2 f(\mathbf{w}; \mathbf{x}_i)}{\partial \mathbf{w}^2}}_{\mathcal{B}(\mathbf{w})} .$$

Assume $\|\boldsymbol{x}_i\| = O(1)$ and $|y_i| = O(1)$ for all $i \in [n]$. For $\mathcal{B}(\mathbf{w}_0)$, by random initialization of weights $\mathbf{w}_0$, with high probability, we have $|f(\mathbf{w}_0; \mathbf{x}_i) - y_i| = O(\log m)$, and $\left\| \frac{\partial^2 f(\mathbf{w}_0; \mathbf{x}_i)}{\partial \mathbf{w}^2} \right\|_2 = \tilde{O}(1/\sqrt{m})$ (Liu et al., 2020; Zhu et al., 2022a) where $m$ denotes the width of the network. Therefore, by the union bound, with high probability, we have $\mathcal{B}(\mathbf{w}_0) = \tilde{O}(1/\sqrt{m})$.

Note that $\lambda_{\max}(\mathcal{A}(\mathbf{w})) = \frac{2}{n} \lambda_{\max}(K(\mathbf{w}))$ for any $\mathbf{w}$. Combining all the bounds together, we have $\left| \lambda_{\max}(H_{\mathcal{L}})(\mathbf{w}_0) - \frac{2}{n} \lambda_{\max}(K)(\mathbf{w}_0) \right| = \tilde{O}(1/\sqrt{m})$. Then we have

$$|\eta_{\text{crit}} - \tilde{\eta}_{\text{crit}}| = \left| \frac{2}{\lambda_{\max}(H_{\mathcal{L}})(\mathbf{w}_0)} - \frac{n}{\lambda_{\max}(K)(\mathbf{w}_0)} \right| = \tilde{O}(1/\sqrt{m})$$

as long as $\lambda_{\max}(K)(\mathbf{w}_0) = \Omega(1)$, which is true with high probability over random initialization for wide networks (Nguyen et al., 2018; Banerjee et al., 2023).

### A.2 EMPIRICAL VALIDATION FOR THE WHOLE TRAINING PROCESS

In this section, we show that Claim 2 still holds if we use $\tilde{\eta}_{\text{crit}}$ which approximates $\eta_{\text{crit}}$. Recall that $\tilde{\eta}_{\text{crit}} = b/\lambda_{\max}(K(\mathbf{w}; X_{\text{batch}}))$ where $b$ is the batch size.

Similar to Figure 4(a), to verify the claim, we compare the sign of $\mathcal{P}\text{Diff}_1$ and $\eta - \tilde{\eta}_{\text{crit}}$ for each batch. In Figure 9, we show the number of events where $\text{sgn}(\mathcal{P}\text{Diff}_1(X_{\text{batch}})) = \text{sgn}(\eta - \tilde{\eta}_{\text{crit}})$ divided by the number of iterations until convergence. We allow a small perturbation, $\epsilon$, in $\eta$ to account for error in estimating $\eta$ due to using finite width models.

In Figure 9(a), we can see that the sign of $\mathcal{P}\text{Diff}_1(\mathbf{f}^t(X_{\text{batch}}))$ and $\eta - \tilde{\eta}_{\text{crit}}(\mathbf{f}^t(X_{\text{batch}}))$ is well matched throughout the training process. Additionally, Figure 9(b) illustrates that $\eta_{\text{crit}}$ is close to $\tilde{\eta}_{\text{crit}}$.

## B $\quad$ ADDITIONAL EXPERIMENTS FOR GD/SGD SPIKES

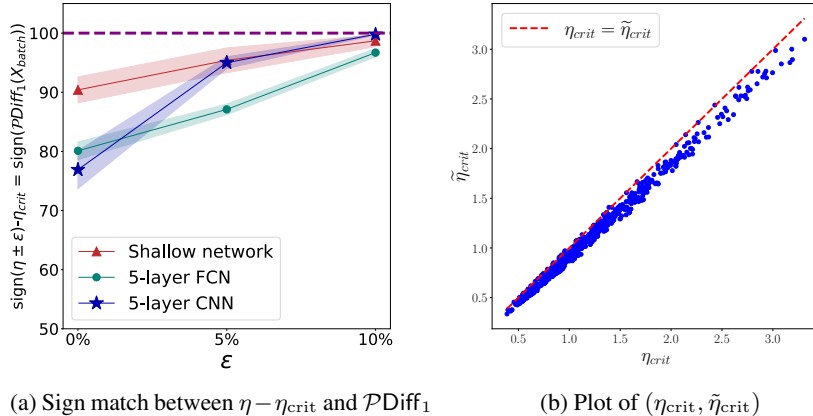

(a) Sign match between $\eta - \eta_{\text{crit}}$ and $\mathcal{P}\text{Diff}_1$

(b) Plot of $(\eta_{\text{crit}}, \tilde{\eta}_{\text{crit}})$

Figure 9: **Validation of $\eta_{\text{crit}} \approx \tilde{\eta}_{\text{crit}}$ during SGD with catapults.** Panel(a): The alignment between $\text{sgn}(\eta - \tilde{\eta}_{\text{crit}})$ and $\text{sgn}(\mathcal{P}\text{Diff}_1^t)$, for which the catapults occur. Panel (b): plot of points $(\eta_{\text{crit}}, \tilde{\eta}_{\text{crit}})$ at each iteration of SGD for the shallow network. All the experiments use CIFAR-10 dataset.

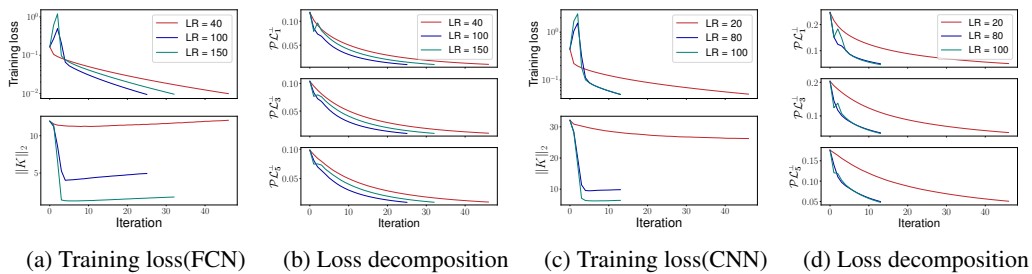

(a) Training loss(FCN)  (b) Loss decomposition  (c) Training loss(CNN)  (d) Loss decomposition

Figure 10: **Catapult dynamics for 5-layer FCN (a-b) and CNN (c-d) on multiclass classification tasks.** Panel (a) and (c) are training loss and the spectral norm of the tangent kernel with different learning rates, and Panel (b) and (d) are training loss decomposed into non-top eigenspace of the tangent kernel, $\mathcal{P}\mathcal{L}_1^{\perp}, \mathcal{P}\mathcal{L}_3^{\perp}$ and $\mathcal{P}\mathcal{L}_5^{\perp}$ (e.g., $\mathcal{P}\mathcal{L}_1^{\perp} = \frac{1}{n}\left\|\mathcal{P}_1^{\perp}(\mathbf{f}(X) - \mathbf{y})\right\|_2^2$). All the networks are trained on a subset of CIFAR-10 with 10 classes. Here the dimension of the eigenspace $s = 1, 3, 5$ refers to $10, 30, 50$ respectively due to the output dimension 10.

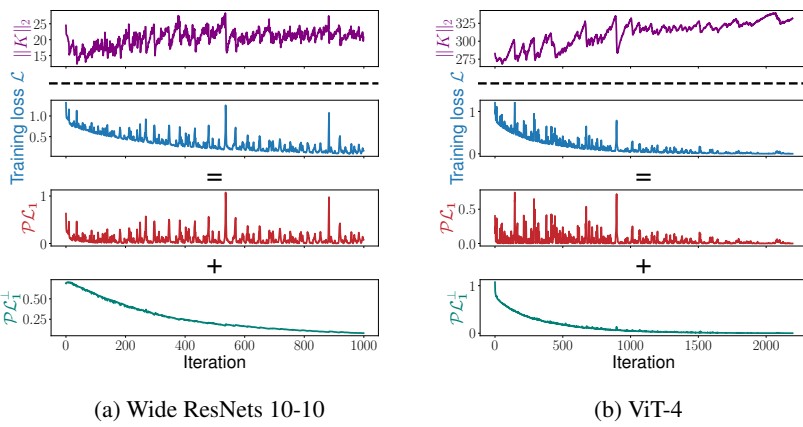

(a) Wide ResNets 10-10  (b) ViT-4

Figure 11: **Cataput dynamics in SGD for modern deep architectures.** The complete versions corresponding to Figure 5(c,d).

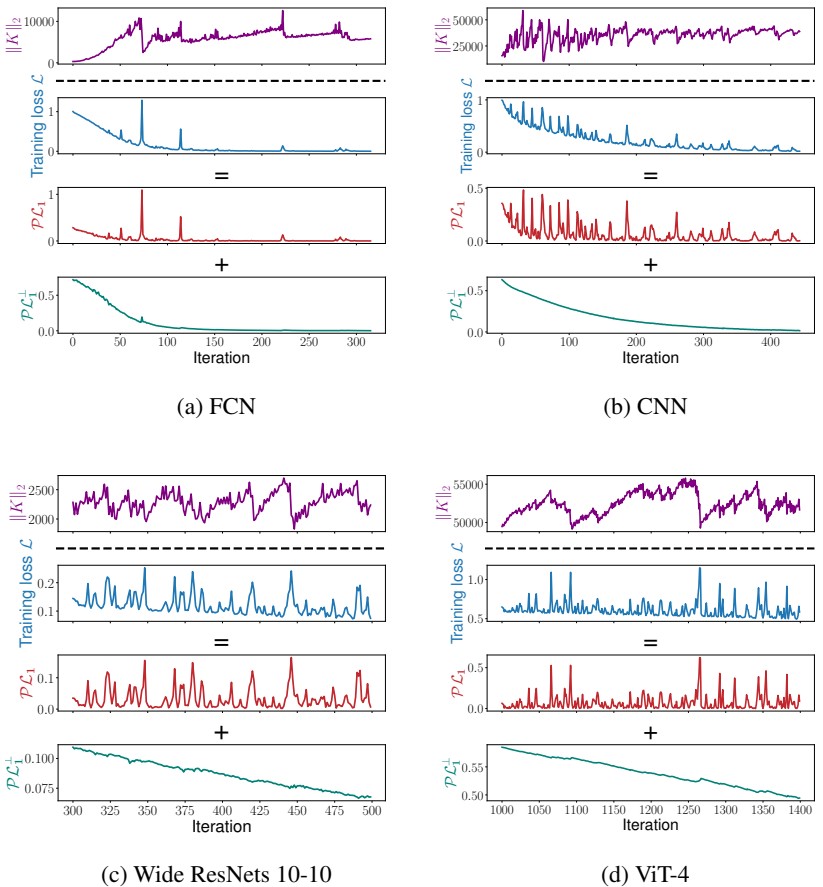

Figure 12: **Cataput dynamics in SGD for modern deep architectures with Pytorch default parameterization.** The tasks are the same with Figure 5 except that we use Pytorch default parameterization. The training loss is decomposed into the top eigenspace of the tangent kernel $\mathcal{PL}_1$ and its complement $\mathcal{PL}_1^\perp$. Here $\mathcal{L} = \mathcal{PL}_1 + \mathcal{PL}_1^\perp$. The training loss and the spectral norm of the tangent kernel correspond to the whole training set.

## C  EXPERIMENTS FOR SGD SPIKES WITH LOGISTIC LOSS

In this work, our focus has been exclusively on the catapult phase phenomenon in the context of mean squared error. This specific focus stems from the fact that the catapult phase was originally observed in training with mean squared error Lewkowycz et al. (2020). Indeed, there is currently no well-established definition of the catapult phenomenon with alternative loss functions. Additionally, it remains unclear whether there is a controlled experiment for the catapult to occur with other loss functions, such as logistic/cross-entropy loss Lewkowycz et al. (2020).

In this section, we show the training dynamics of neural networks with logistic loss. We consider the same experimental setting with Figure 5(c,d), with the only modification being the substitution of mean squared error for logistic loss, i.e., $\mathcal{L}(\mathbf{w}; (X, y)) = \frac{1}{n}\sum_{i=1}^{n} \log\left(1 + e^{-f(\mathbf{w};\boldsymbol{x}_i)y_i}\right)$. Interestingly, we observe that there are spikes in the training loss using SGD, and the spectral norm of the tangent kernel also decreases when there is a loss spike. See Figure 15. This observation aligns with our findings related to mean squared error, as discussed in Section 3.3. It implies that the catapult phase phenomenon may also occur with logistic loss. However, additional evidence is required for conclusive verification.

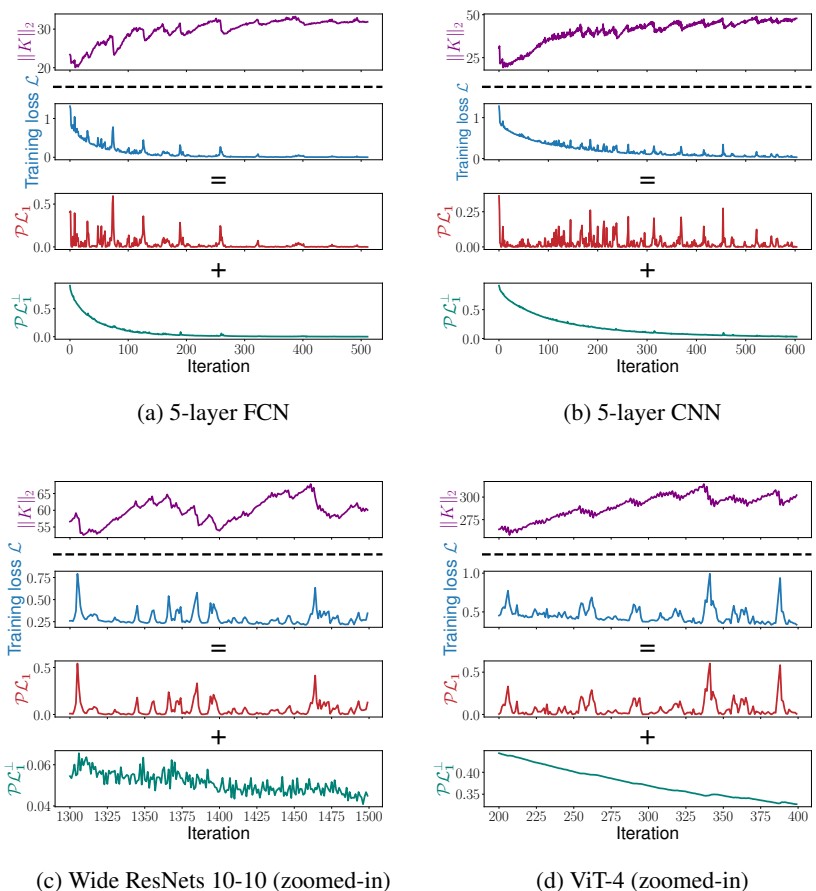

(a) 5-layer FCN

(b) 5-layer CNN

(c) Wide ResNets 10-10 (zoomed-in)

(d) ViT-4 (zoomed-in)

Figure 13: **Cataput dynamics in SGD for modern deep architectures on 2-class SVHN.** The tasks are the same with Figure 5 except that we train the neural networks on a subset of SVHN dataset. The training loss is decomposed into the top eigenspace of the tangent kernel $\mathcal{PL}_1$ and its complement $\mathcal{PL}_1^\perp$. Here $\mathcal{L} = \mathcal{PL}_1 + \mathcal{PL}_1^\perp$. The training loss and the spectral norm of the tangent kernel correspond to the whole training set.

## D    ADDITIONAL EXPERIMENTS FOR FEATURE LEARNING IN GD

We present the validation loss/error for the tasks corresponding to Figure 6. The learning rate is repeatedly increased during training to generate multiple catapults.

We compare the performance of networks exhibiting multiple catapults with those initialized using small initialization scheme, i.e., each weight is sampled i.i.d. from $\mathcal{N}(0, \sigma^2)$ with $\sigma = 0.1$. This is in contrast to the NTK parameterization where we use $\sigma = 1$. We can see that small initialization achieves the smallest test loss/error as well as the best AGOP alignment, which indicates that learning AGOP correlates strongly with the test performance.

For the Rank-2 regression task, we visualize the AGOP in the following Figure 18, where we can see that the features are learned better, i.e., closer to the True AGOP, with a greater number of catapults.

## E    ADDITIONAL EXPERIMENTS FOR FEATURE LEARNING IN SGD

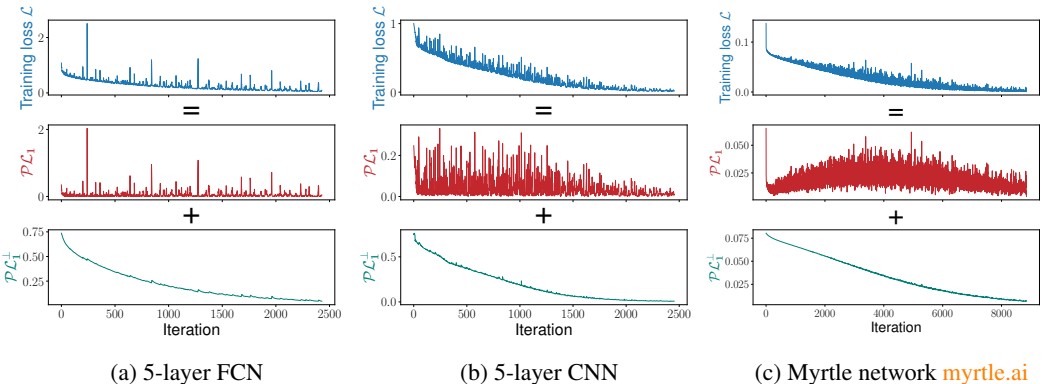

(a) 5-layer FCN     (b) 5-layer CNN     (c) Myrtle network myrtle.ai

Figure 14: **Catapult dynamics in SGD for large datasets (Panel (a) and (b)) and multi-class classification problems (Panel(c)).** Panel(a,b): The networks are trained on $5,000$ data points from CIFAR-2. Panel(c): The network is trained on $128$ points from CIFAR-10. The training loss is decomposed into the top eigenspace of the tangent kernel $\mathcal{PL}_1$ and its complement $\mathcal{PL}_1^{\perp}$. Here $\mathcal{L} = \mathcal{PL}_1 + \mathcal{PL}_1^{\perp}$. The training loss corresponds to the whole training set.

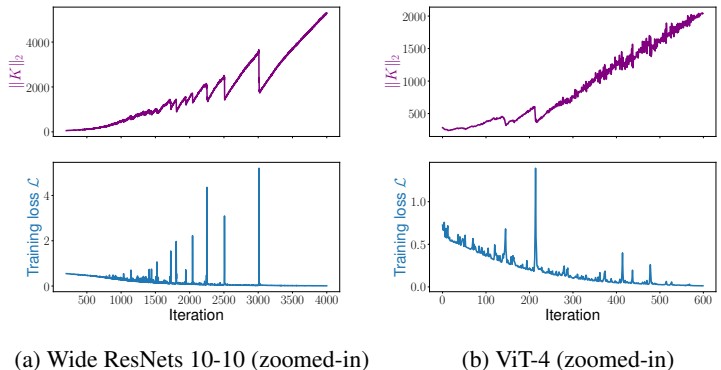

(a) Wide ResNets 10-10 (zoomed-in)     (b) ViT-4 (zoomed-in)

Figure 15: **Catapult dynamics in SGD with logistic loss.** The experimental setting is the same as Figure 5(c,d), with the only modification being the substitution of mean squared error for logistic loss.

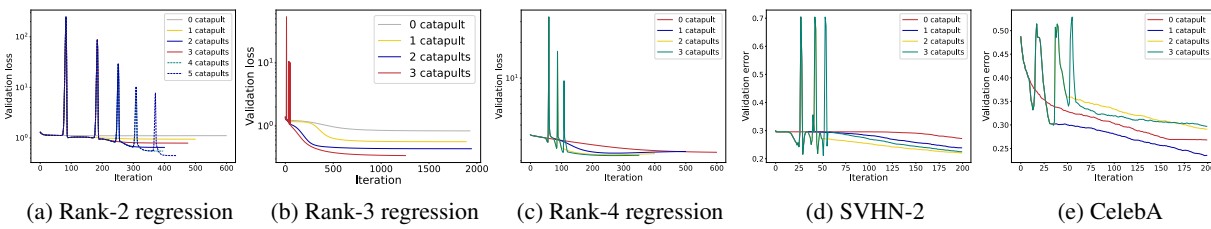

(a) Rank-2 regression    (b) Rank-3 regression    (c) Rank-4 regression    (d) SVHN-2    (e) CelebA

Figure 16: **Validation loss/error of multiple catapults in GD corresponding to** Figure 6. Panel(d)&(e) only present first 200 iterations.

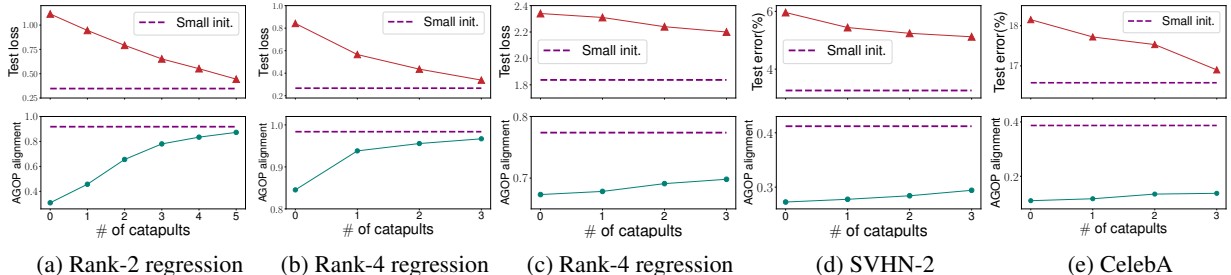

(a) Rank-2 regression    (b) Rank-4 regression    (c) Rank-4 regression    (d) SVHN-2    (e) CelebA

Figure 17: **Multiple catapults in GD compared to the small initialization scheme.** We train a 2-layer FCN in Panel(a), a 4-layer FCN in Panel(b,c,e) and a 5-layer CNN in Panel(d). For small initialization, each weight parameter is i.i.d. from $\mathcal{N}(0, \sigma^2)$ with $\sigma = 0.1$. The experimental setup is the same as Figure 6.

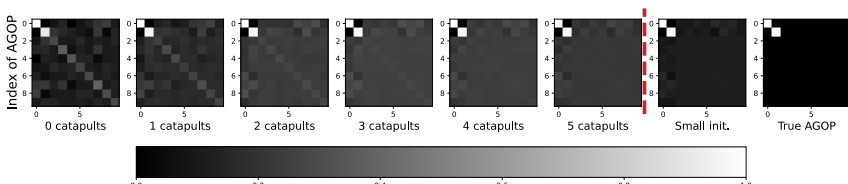

Figure 18: **Visualization of AGOP for rank-2 regression task.** All pixels are normalized to the range $[0, 1]$ and the top 10 rows and columns of the AGOP are plotted.

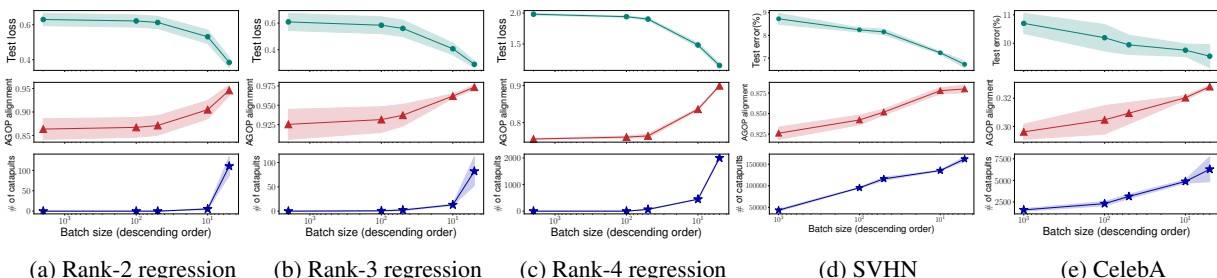

(a) Rank-2 regression    (b) Rank-3 regression    (c) Rank-4 regression    (d) SVHN    (e) CelebA

Figure 19: **Similarity between AGOP alignment and test performance with Pytorch default parameterization.** The tasks are the same with Figure 7 except that we use Pytorch default parameterization.

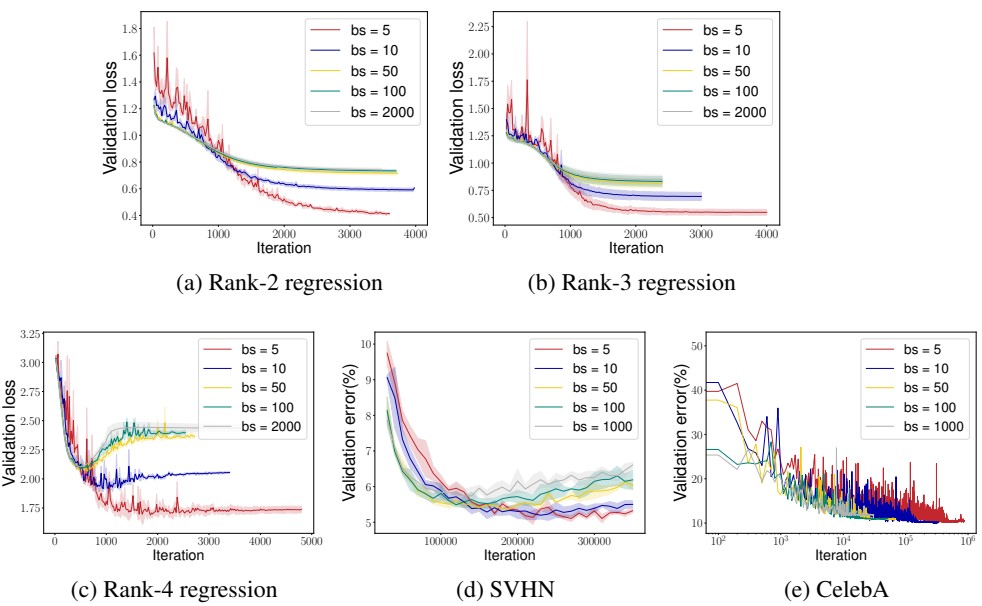

Figure 20: **Validation loss/error corresponding to** Figure 7. Panel(c) presents the validation error from iteration 4000.

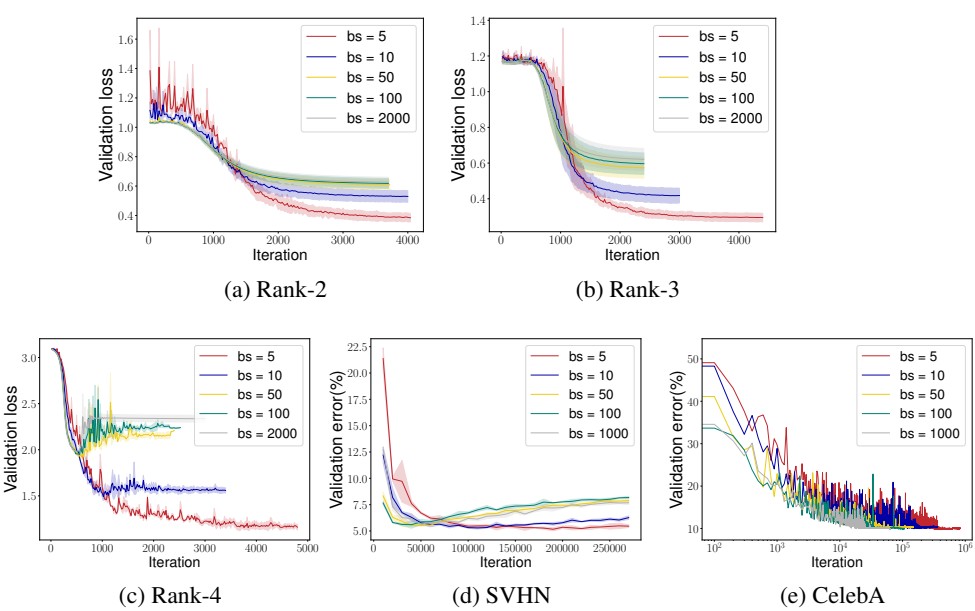

Figure 21: **Validation loss/error with Pytorch default parameterization corresponding to** Figure 19. Panel(c) presents the validation error from iteration 2000.

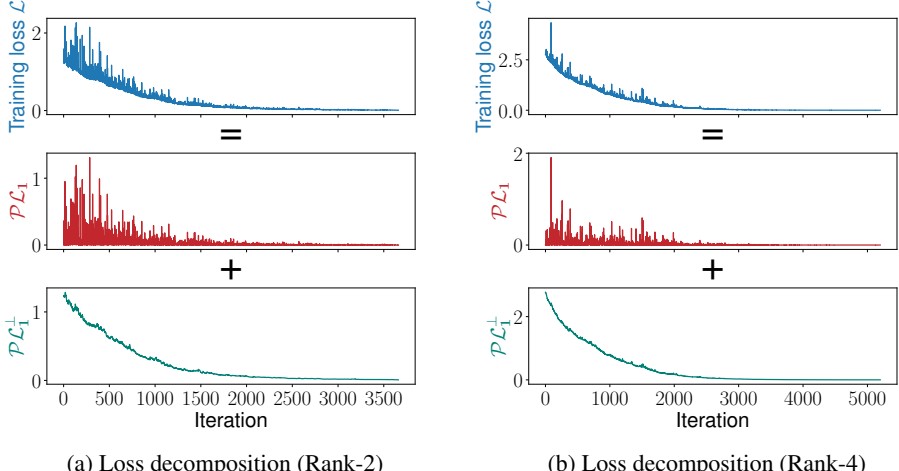

(a) Loss decomposition (Rank-2)  (b) Loss decomposition (Rank-4)

Figure 22: **Verification of catapult dynamics: loss decomposition of Rank-2 and Rank-4 regression tasks corresponding to** Figure 7. The training loss is decomposed into the top eigenspace of the tangent kernel $\mathcal{PL}_1$ and its complement $\mathcal{PL}_1^\perp$. Here $\mathcal{L} = \mathcal{PL}_1 + \mathcal{PL}_1^\perp$.

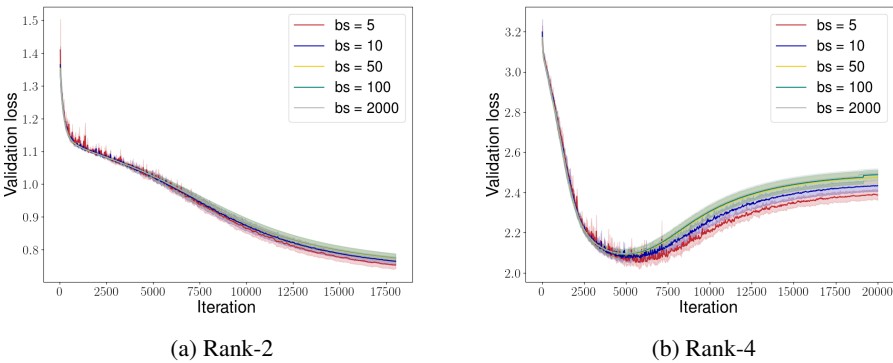

(a) Rank-2  (b) Rank-4

Figure 23: **The networks are trained with a small learning rate corresponding to** Figure 20a &b. With a small learning rate, no catapults occur during training, hence the effect of batch size is not sigfinicant.

## F  EXPERIMENTAL DETAILS

For all the networks considered in this paper, we use ReLU activation functions. Except for the experiments in Figure 12 and 19, we parameterize the network by NTK parameterization.

**NTK parameterization.** Given a neural network with NTK parameterization, all the trainable weight parameters are i.i.d. from $\mathcal{N}(0,1)$. For a fully connected layer, it takes the form $f^{\ell+1} = \text{ReLU}\left(\frac{1}{\sqrt{m_\ell}}W^\ell f^\ell + b^\ell\right)$ where $W^\ell \in \mathbb{R}^{m_{\ell+1}\times m_\ell}, f^\ell \in \mathbb{R}^{m_\ell}, b^\ell \in \mathbb{R}^{m_{\ell+1}}$. For a convolutional layer, it takes the form $f^{\ell+1}_{i,j,k} = \text{ReLU}\left(\frac{1}{\sqrt{m_\ell s^2}}\sum_{p=0}^{\lceil\frac{s+1}{2}\rceil}\sum_{q=0}^{\lceil\frac{s+1}{2}\rceil}\sum_{o=1}^{m_\ell}W^\ell_{p,q,o,k}f^\ell_{i-\lceil\frac{s-1}{2}\rceil,j-\lceil\frac{s-1}{2}\rceil,o} + b^\ell_k\right)$, where $W^\ell \in \mathbb{R}^{s\times s\times m_\ell\times m_{\ell+1}}, f^\ell \in \mathbb{R}^{h\times w\times m_\ell}, b^\ell \in \mathbb{R}^{m_{\ell+1}}$. Note that $s$ is the filter size and we assume the stride to be 1 in this case. For $f^\ell$ with negative indices, we let it be 0, i.e., zero padding. For the output layer, we use a linear layer without activation functions.

**Dataset.** For the synthetic datasets, we generate data $\{(\boldsymbol{x}_i, y_i)\}_{i=1}^{n}$ by i.i.d. $\boldsymbol{x}_i \sim \mathcal{N}(0, I_{100})$ and $y_i = f^*(\boldsymbol{x}) + \epsilon$ with $\epsilon \sim \mathcal{N}(0, 0.1^2)$. For two real-world datasets, we consider a subset of CelebA dataset with glasses as the label and the Street View House Numbers (SVHN) dataset. Due to computational limitations with GD, for some tasks, we select two classes of SVHN dataset (number 0 and 2), which we refer to as SVHN-2.

**True AGOP.** Note that for these low-rank polynomial regression tasks, we know the target functions hence we can compute the true AGOP by $G^* = \frac{1}{n} \sum_{i=1}^{n} \frac{\partial f^*}{\partial \boldsymbol{x}_i} \frac{\partial f^*}{\partial \boldsymbol{x}_i}^{T}$. For real-world datasets, we estimate the true AGOP by using one of the state-of-the-art models that achieve high test accuracy.

In the following, we provide the detailed experimental setup for each experiment. Note that in the classification tasks, i.e. CeleA and SVHN datasets, the test error refers to the classification error on the test split.

### F.1 FIGURES IN SECTION 3.1

**Figure 2:** We use a 2-class subset of CIFAR-10 dataset Krizhevsky et al. (2009) (class 7 and class 9) and randomly select 128 data points out of it. For the network architectures, we use a 5-layer FCN with width 1024 and 5-layer CNN with 512 channels per layer. For CNN, we flatten the image into a one-dimensional vector before the last fully connected layer.

### F.2 FIGURES IN SECTION 3.2

**Figure 3:** We use the same training tasks as in Figure 2. For FCN, we start with lr = 6 and we increase the learning rate to $[10, 15]$ at iteration $[15, 60]$. For CNN, we start with lr = 8 and we increase the learning rate to $[15, 20]$ at iteration $[10, 40]$.

### F.3 FIGURES IN SECTION 3.3

**Figure 4:** For the 5-layer FCN and 5-layer CNN, we use the same network architectures as in Figure 2. For the shallow network, we use a 2-layer FCN with width 1024. We train the model on 128 datapoints from CIFAR 2 using SGD with batch size 32. The learning rates for 5-layer FCN, 5-layer CNN and the shallow network are $6, 8, 0.8$ respectively. We stop training when the training loss is less than $10^{-3}$.

**Figure 5:** The 5-layer FCN and CNN are the same as in Figure 2. And we use the standard Wide ResNets and ViT architectures. All the models are trained on 128 data points from CIFAR-2 using SGD with batch size 32.

### F.4 FIGURES IN SECTION 4

**Figure 6:** For rank-2 task, we train a 2-layer FCN with width 1024. The size of the training set, testing set and validation set are $2000, 5000$ and $5000$ respectively.

For rank-3 task, rank-4 task and CelebA tasks, we train a 4-layer FCN with width 256. The size of the training set, testing set and validation set are $1000, 5000$ and $5000$ respectively.

For SVHN-2 tasks, we train a 5-layer CNN with width 256. We select class 0 and class 2 out of the full SVHN datasets as SVHN-2. The size of the training set, testing set and validation set are $1000, 5000$ and $5000$ respectively.

We increase the learning rate during training. For Rank-2 task, we increase the learning rate to $[8, 16, 30, 50, 75, 80]$ at iteration $[50, 150, 220, 280, 350, 400]$. For Rank-3 task, we increase the learning rate to $[40, 100, 150]$ at iteration $[20, 60, 80]$. For Rank-4 task, we increase the learning rate to $[15, 40, 60]$ at iteration $[50, 75, 110]$. For SVHN-2 task, we increase the learning rate to $[30, 60, 90]$ at iteration $[10, 35, 50]$. For CelebA task, we increase the learning rate to $[40, 70, 100]$ at iteration $[10, 35, 50]$. We decay the learning rate if necessary after the catapult to avoid extra catapults until the end of training.

**Figure 7:** For both Rank-2 and Rank-4 tasks, we let the size of training set, testing set and validation set be $2000, 5000$ and $5000$. For the SVHN task, we train the full SVHN using the 5-layer Myrtle network. For the CelebA task, we train the full 2-class CelebA dataset with glasses feature using 4-layer FCN with width 256. To obtain the true AGOP , we use one of the SOTA models (WideResNet 16-2) which achieves $97.2\%$ test accuracy on SVHN and 5-layer Myrtle network which achieves $95.7\%$ test accuracy on CelebA.

We use the same learning rate across batch sizes for each task. The learning rate is chosen as $\frac{1}{2}\eta_{\mathrm{crit}}$ corresponding to the whole training set. For SVHN and CelebA tasks, we estimate $\eta_{\mathrm{crit}}$ using a subset with size 5000 of the whole training set. For all tasks, we stop training when the training loss is less than $10^{-3}$. We report the average of 3 independent runs.

**Figure 8:** We use the same network architectures and training/validation/testing sets as in Figure 6.

For all the tasks, except for GD, all the optimizers use a mini-batch size 100.

We stop training when the training loss is less than $10^{-3}$. We report the average of 3 independent runs.

For the rank-2 task and rank-4 task, we know the target function hence we can analytically compute the exact true AGOP . For SVHN-2 task and CelebA task, to estimate the true AGOP , we use one of the SOTA models, Myrtle-5 which achieves $98.4\%$ test accuracy on two-class SVHN dataset and $95.7\%$ test accuracy on CelebA dataset.

The following table is the learning rate we choose for the experiments:

| Task | SGD | GD | SGD+M | Adadelta | Adagrad | RMSprop | Adam |
|---|---|---|---|---|---|---|---|
| Rank-2 | 2.0 | 2.0 | 2.0 | 2.0 | 0.1 | $10^{-2}$ | $10^{-2}$ |
| Rank-3 | 2.0 | 2.0 | 2.0 | 2.0 | $10^{-2}$ | $10^{-2}$ | $10^{-3}$ |
| Rank-4 | 1.0 | 1.0 | 1.0 | 1.0 | $5 \times 10^{-3}$ | $10^{-3}$ | $10^{-3}$ |
| SVHN-2 | 5.0 | 5.0 | 5.0 | 5.0 | $5 \times 10^{-3}$ | $10^{-4}$ | $10^{-3}$ |
| CelebA | 10.0 | 10.0 | 10.0 | 10.0 | $5 \times 10^{-3}$ | $10^{-3}$ | $10^{-3}$ |

Table 2: **Choice of learning rates for Figure 8.**

The experiment is to demonstrate the correlation between AGOP alignment and test performance. For this reason, we did not fine-tune the learning rate to achieve the best test performance.

### F.5 FIGURES IN APPENDIX B

**Figure 10:** We use the same network architectures as in Figure 2 and we train 128 data point from CIFAR-10.

**Figure 13:** We use the same setting as Figure 5, except we train the networks on 128 data points from SVHN-2(number 0 and 2).

**Figure 14:** For panel(a) and panel(b), we train the same 5-layer FCN and CNN as in Figure 2 and on 5,000 data points from CIFAR-2. For panel(c), we train a 5-layer Myrtle network on 128 points from CIFAR-10.

### F.6 FIGURES IN APPENDIX E

**Figure 19:** We use the same setup with Figure 7 except that all the networks are parameterized with Pytorch default parameterization. The learning rates are 0.01, 0.01, 0.05 and 1.0 for each task.