# OpenReview forum: "Catapults in SGD: spikes in the training loss and their impact on generalization through feature learning"
_ICLR.cc/2024/Conference — Submitted to ICLR 2024_

### Official Review · Reviewer_QEoS · 2023-10-21

**Soundness:** 2 fair
**Presentation:** 2 fair
**Contribution:** 3 good
**Rating:** 5
**Confidence:** 4

**Summary:**

This work studies the catapult phenomenon of (S)GD, which happens when we use a large enough step size that results in a spike in the training loss. This is mainly an empirical work, and they focus on various models under the MSE loss. Through extensive experiments, they show empirically that whenever catapults occur, the loss also spikes in the top eigenspace of the neural tangent kernel (NTK) matrix. Furthermore, they show that more catapults corresponds to an increased alignment between the trained model and the groundtruth model. This alignment is measured in terms of the AGOP of the respective models, where AGOP stands for the average gradient outer product, a matrix known to capture learned features. From this observation, they claim that catapults lead to better generalization.

**Strengths:**

The catapult phenomenon studied is interesting and deserves attention in the ICLR community. Although it has been observed and studied in the past, this paper provides new insights into how catapults are decomposed in the eigenspace of the NTK, and how they correlate with better generalization. These are novel directions to study this problem. The connection between catapults and AGOP alignment metric is also original and interesting.

**Weaknesses:**

Although the experiments are quite extensive and the observations are interesting, the paper in its current state is somewhat weak in both impact and presentation. In summary, I'm concerned about how much of the results can carry over to a realistic deep learning setting, where the cross entropy loss is most often used for classification, whereas in this work only the MSE loss is considered. There are many observations, but there lacks a coherent message. The language used in this paper can be inaccurate and handwavy, and I'm afraid the experiment details provided is insufficient for easy reproducibility. The presentation also needs some improvement.


### Major concerns

- Although I'm aware that Lewkowycz et al. (2020) also studied catapults only for the squared loss, it's still concerning that this work also does this same. Given that the logistic/cross-entropy loss is so often used in deep learning, at least some of the experiments should involve this setting. Especially since many of the tasks in this work is on classification, it's unrealistic to use the squared loss as it's extremely rare to train a classification model this way. This makes me wonder whether one can observe or induce catapults at all when using the logistic or cross-entropy loss, and whether the observations and claims in this paper can be carried in this more realistic setting.

- The authors repeatedly emphasize that they show "catapults mainly occur in the subspace spanned by the top eigenvectors of the tangent kernel". After reading the entire paper, I'm still confused about this observation. Yes, the experiments show that this indeed seems to happen. But the question is: so what? Should I be surprised about it? The top eigenspace of the NTK corresponds to the examples in the training set that matter the most in terms of prediction, so they are likely the "support vectors". So if catapults were to occur, it's not surprising that the projection of the residual also spikes for these examples. I think more elaboration about this observation is needed. Why is it significant, and why should we care about the NTK matrix when it comes to catapults? This part of the paper also seems to be somewhat disconnected from the latter part - it's unclear how "catapults show up in the top eigenspace of the NTK" is related to the AGOP alignment. In fact, upon first reading of this observation, it's not immediately clear what it means or what it implies. Further elaboration of this point is need to strengthen the paper, otherwise it just seems like a standalone observation that may or may not be significant.

- The paper claims to connect three phenomena in deep learning, the third one being "small batch SGD leads to better generalization" (paraphrased third point at the bottom of page 1). As far as I'm aware, the paper by Geiping et al., 2022 [Stochastic Training is Not Necessary for Generalization](https://arxiv.org/abs/2109.14119) says that's not always the case if you tune well. It seems like in Figure 7, 18-20, only one step size is given for each task (page 22), so I'm assuming they are the same across batch sizes as well. It's unclear how these step sizes are selected and why they are the same for all batch sizes. I'm a bit skeptical about this setup, especially given the results in Geiping et al., 2022.

- Claim 2 is about changes in the (projected) batch loss, but the experiments on SGD all show the full loss. Is Claim 2 really significant if what we truly care about is the full loss?

- Page 2: "This explains why the loss drops quickly to pre-spike levels from the peak of the spike". First, I don't see how catapult in the eigenspace explains this. Second, I find this entire sentence vague and handwavy. It's unclear what the authors mean by pre-spike levels. Is it the loss level at the iteration right before the spike occurs, or 50 iterations before the spike occurs? What drop rate is considered quick? Quick compared to what?

- More experiment details should be given and in a clear manner, rather than having them scattered around throughout the main text and the appendix.
	- For instance, the paper mentions that the NTK parameterization is used in many places, but what exactly is this parameterization is unclear, especially to readers who may not be familiar with it. I recommend that the authors explicitly describe what the model architecture is, what the initializations are (crucial to the NTK parameterization), etc. More importantly, why is the NTK parameterization used? Is it crucial that you must use this to produce the results?
	- Another one is the step size used for Figure 3: yes, we can sort of read off the step size from the LR figure, but the exact numbers are not given. How did you choose those numbers? It's said in the next page that it starts small $\eta<\eta_{\mathrm{crit}}\approx\eta/\lambda_{\max}(K)$ and increases to $\eta>\eta_{\mathrm{crit}}$, but how exactly is it achieved? What is $\approx$? Is the step size increased by a multiplicative factor? I don't think this information is explicitly provided in the paper, and I apologize in advance if I had missed it somewhere.

- The related work section needs more work, in the sense that it currently presents a laundry list of related works, but how they are related to each other and to the current work is inadequately discussed. Specifically, at the end of the **Catapult phase** paragraph, instead of explaining what catapult is, which was already done in the intro, here the authors should elaborate more on what these works have found, what setup was different, how their results relate to the results of the current work, and so on. Similar improvements can be made to the **EoS paragraph.

**Questions:**

- Why does the critical step size only depend on the initialization and not varying throughout training?
- Paragraph under Claim 1: shouldn't it be $\mathcal{P}\mathcal{L}_n$ instead of  $\mathcal{P}\mathcal{L}_5$ that corresponds to the the spike in the training loss? Is $\mathcal{P}\mathcal{L}_5$ even plotted somewhere that I missed?
- **Choice of top eigenspace dimension $s$**:
	- The first sentence after this  - I'm confused by what this sentence is trying to convey, especially the second part. What linear dynamics?
	- So you expect to need a larger $s$, and turns out a small $s$ is sufficient to produce the catapults in the top-$s$ eigenspace. Do you have an explanation to why this contradicts to what you expect? My guess is that $s$ corresponds to the support vectors in some sense, but I'm also uncertain. Is there a way to verify this?
- Eq (6): $\mathbf{f}(X_{\mathrm{batch}})$ has the same length as the number of examples in the batch. But the projection operator acts on prediction vectors of length $n$. Is something off here or am I misunderstanding?
- Do you need to recompute $\eta_{\mathrm{crit}}(X_{\mathrm{batch}})$ for each batch? How does it work?
- I think more motivation for "AGOP alignment is a good measure for generalization" can be given. As a simple example, one can consider the classic low-rank matrix sensing problem
$$
L(U) = \frac{1}{2n}\sum_{i=1}^n ( \langle A_i,UU^\top \rangle - y_i )^2
$$
where the labels are generated via $y_i= \langle A_i, X^* \rangle$ for some positive semidefinite low-rank groundtruth $X^*$. Since $G$ and $G^*$ can be derived in closed form, does high AGOP alignment imply high generalization in terms of $\Vert UU^\top - X^*\Vert_F$?


### Minor suggestions
- Latin words such  as *a priori* should be italicized
- Figure 1 caption: training loss "of" SGD -> training loss "when optimized using" SGD
- Table 2: please use $0.01$ or $10^{-2}$ instead of 1e-2 to present hyperparameters used.
- Page 2: sentence starting with "Namely, we project the residual..."
	- Up to this point, the paper has not mentioned that it only looks at the MSE loss, so readers who expect the authors to also study the logistic/cross-entropy loss may be confused why you are projecting the residual.
- The experimental details section (E) can be improved in terms of writing: there are many dangling paragraphs with only one or two sentences, which break the flow of the writing and reads more like bullet points that don't belong together.
- Section E, 3rd paragraph: "the test error refers to the classification error" -> "the test error refers to the classification error on the test split"
- I may be overly fussy about this or have interpreted something incorrectly, but in the related work section, the authors say that "In Cohen et al. (2020)", it was conjectured that at EoS, there are numerous catapults". To me it's more like Cohen et al. "observed" numerous catapults at EoS, but conjectured that EoS and catapults have the same underlying cause.
- Page 4 second paragraph:
	- "the tangent kernel $K=$... and $H_{\mathcal{L}}=$..." -> "the tangent kernel is given by $K=$... and the Hessian is $H_{\mathcal{L}}=$..."
	- "As wide networks are close to their linear approximations" -> "As wide networks are close to their linear approximations when in the NTK regime"
- When defining the AGOP alignment in page 4, the "true model" $f^*$ should be defined more rigorously. Since $w$ is the set of parameters for any given model throughout the paper, one could say something along the lines of "let $w$ be the parameters for the trained model, and $w^*$ be the parameters minimizing the population objective or a reference model that achieves SoTA accuracy on a test set". At this point, the reader can be confused to what exactly is the true model, as with an arbitrary dataset and model architecture, you don't always have access to the data-generating model.
- First sentence in Section 3.1: "eigenvalues" -> "eigenvectors"
- It's better to move Definition 2 to the beginning of Section 4, where AGOP alignment first show up in experiments and discussion.

---

> ### Author Response · Authors · 2023-11-19
>
> We thank the reviewer for the detailed and insightful comments. We will address your concerns and questions below.
>
> *Concern 1.a: Given that the logistic/cross-entropy loss is so often used in deep learning, at least some of the experiments should involve this setting. Especially since many of the tasks in this work is on classification, it's unrealistic to use the squared loss as it's extremely rare to train a classification model this way....whether one can observe or induce catapults at all when using the logistic or cross-entropy loss, and whether the observations and claims in this paper can be carried in this more realistic setting.*
>
> To the best of our knowledge, there is no clear definition of catapults with other loss functions.  Indeed, all previous work on catapults has focused on the squared loss (see [1]).  We added evidence in Figure 15 that is suggestive of catapults under cross-entropy loss but a thorough investigation of this phenomenon is beyond the scope of our work.  We additionally note that the squared loss is competitive with cross-entropy across a variety of different classification tasks and architectures (see extensive experimental results in [2, 3]).
>
> 1.  A. Lewkowycz, et al. "The large learning rate phase of deep learning: the catapult mechanism." arXiv preprint arXiv:2003.02218 (2020).
>
> 2. L. Hui, M. Belkin "Evaluation of neural architectures trained with square loss vs cross-entropy in classification tasks." ICLR 2021.
>
> 3. K. Janocha, and C. Wojciech M.  "On loss functions for deep neural networks in classification." arXiv preprint arXiv:1702.05659 (2017).
>
> *Concern 2.a: Why is the observation "catapults mainly occur in the subspace spanned by the top eigenvectors of the tangent kernel" significant? The top eigenspace of the NTK corresponds to the examples in the training set that matter the most in terms of prediction, so they are likely the "support vectors". So if catapults were to occur, it's not surprising that the projection of the residual also spikes for these examples....more elaboration about this observation is needed.*
>
> We are the first to observe that catapults mainly occur in the subspace spanned by the top eigenvectors of the tangent kernel. In our opinion, it is surprising as the catapult dynamics of complicated neural networks, including Wide ResNets and Vision Transformer, indeed occur in a low dimensional subspace.
>
> We are not sure we understood your comment about the ''support vectors''. Could you please clarify how you see the connection between top eigenvectors of the NTK and the individual training examples?
>
> *Concern 2.b: Why should we care about the NTK matrix when it comes to catapults?*
>
> Catapults are related to the loss function, which is controlled by the Hessian. The NTK and the Hessian of the loss are closely related. This can be seen as follows: for MSE  $\mathcal{L}(W;X) = \frac{1}{n}\sum_{i=1}^n (f(W;x_i)-y_i)^2$, we can compute its $H_\mathcal{L}$ by the chain rule:
> \begin{align}
>  H_\mathcal{L}(w) = \frac{2}{n}(A(w) +B(w)),
> \end{align}
> where $A(w) = \sum_{i=1}^n\left(\frac{\partial f(w;x_i)}{\partial w}\right)^T \frac{\partial f(w;x_i)}{\partial w}$, and $B(w) = \sum_{i=1}^n  (f(w;x_i)-y_i)\frac{\partial^2 f(w;x_i)}{\partial w^2}$.
>
> Here $A(w)$ shares the same eigenvalue with NTK with extra $p-n$ $0$ eigenvalues ($w\in\mathbb{R}^p$). $||B(w)||$ is small due to $|f_i-y_i| = O(1)$ and $||\frac{\partial^2 f(w;x_i)}{\partial w^2}|| = \tilde{O}(1/\sqrt{m})$ during training in the kernel regime [1].  Therefore, the spectrum of the tangent kernel is close to the spectrum of $H_\mathcal{L}(w)$ in the kernel regime.  As the critical learning rate is determined by the top eigenvalue of the Hessian, we can estimate the critical learning rate by NTK which greatly improves the time efficiency as $p\gg n$. This approximation of critical learning rates is validated in catapult dynamics as well. Please see Appendix A for experiments.
>
> Furthermore, the catapult was originally characterized as a spike in the training loss and a decrease in the spectral norm (i.e., largest eigenvalue) of the NTK [Lewkowycz et al. 2020].
>
> [1]C. Liu, et al. "On the linearity of large non-linear models: when and why the tangent kernel is constant." Advances in Neural Information Processing Systems 33 (2020): 15954-15964.

---

> ### Author Response · Authors · 2023-11-19
>
> *Concern 2.c: This part of the paper also seems to be somewhat disconnected from the latter part - it's unclear how "catapults show up in the top eigenspace of the NTK" is related to the AGOP alignment.*
>
> The observation that "catapults show up in the top eigenspace of the NTK" is not directly related to the AGOP alignment in the paper. The contributions of our work include two areas: optimization, where we propose an explanation for the spikes in the training loss using SGD;  and generalization, where we explain how catapults improve generalization. The observation that catapults show up in the top eigenspace of the NTK serves as evidence that the loss spikes in SGD are catapults (contribution in optimization). AGOP alignment is the feature learning that explains how catapults improve generalization (contribution in generalization).  Both contributions are about the impact of catapults in neural networks.
>
> *Concern 3.a: "small batch SGD leads to better generalization"  is not always the case given the paper by Geiping et al., 2022*
>
> Thanks for the interesting reference. It is certainly widely believed that small batch sizes generally lead to better generalization[1,2,3,4].  We have no specific opinion on the prevalence of this behavior but rather focus on showing how catapult can lead to such behavior.
>
> 1. I. Kandel, and C. Mauro. "The effect of batch size on the generalizability of the convolutional neural networks on a histopathology dataset." ICT express 6.4 (2020): 312-315.
> 2. D. Masters, and L. Carlo. "Revisiting small batch training for deep neural networks." arXiv preprint arXiv:1804.07612 (2018).
> 3. N. Keskar S., et al. "On large-batch training for deep learning: Generalization gap and sharp minima." arXiv preprint arXiv:1609.04836 (2016).
> 4. S. Smith L., et al. "On the Origin of Implicit Regularization in Stochastic Gradient Descent." International Conference on Learning Representations. 2020.
>
> *Concern 3.b: It seems like in Figure 7, 18-20, only one step size is given for each task (page 22), so I'm assuming they are the same across batch sizes as well. It's unclear how these step sizes are selected and why they are the same for all batch sizes. I'm a bit skeptical about this setup, especially given the results in Geiping et al., 2022.*
>
> We used the same step size across batch sizes for each task in Figure 7. The step size is chosen as $\frac{1}{2}\eta_{crit}$ corresponding to the whole training set. For large datasets e.g., SVHN, we estimate $\eta_{crit}$ using a subset of the whole training set. We have clarified the details in the appendix.
>
> The reason we use the same learning rate for different batch sizes is as follows. Consider the update rule of SGD: $w_{t+1} = w_t - \frac{\eta}{b} \frac{\partial }{\partial w} \sum_{j\in X_{batch}}\ell_j(w_t) $ where  $\ell_j(w_t) = (f(w_t;x_j)-y_j)^2$ and $b = |X_{batch}|$.  If we assume $(x_j,y_j)$ are i.i.d., then $\mathbb{E} \left[\frac{1}{b}\frac{\partial }{\partial w} \sum_{j\in X_{batch}}\ell_j(w_t)\right] = \mathbb{E} \left[\frac{\partial }{\partial w} \ell_j(w_t)\right] $. Note that a larger batch size will make the gradient closer to the expectation, and a smaller batch size can have a higher variance in the gradient. To ensure the magnitude of the norm of weight update is similar, we use the same learning rate for different batches.
>
> *Concern 4: Claim 2 is about changes in the (projected) batch loss, but the experiments on SGD all show the full loss. Is Claim 2 really significant if what we truly care about is the full loss?*
>
> Claim 2 shows that the mechanism of the catapults in GD, that it occurs when the learning rate is larger than the critical one, also holds in SGD. Since the gradient of the loss is calculated based on the mini-batch in SGD, the mechanism also manifests in the change of the batch loss. Together with the observation on the behavior of the training loss and the NTK, it serves as a strong support that catapults also occur in SGD.

---

> > ### Comment · Reviewer_QEoS · 2023-11-19
> >
> > >  catapults show up in the top eigenspace of the NTK serves as evidence that the loss spikes in SGD are catapults
> >
> > I still find this quite vague --- aren't catapults just spikes in the training loss, i.e., non-monotonically decreasing training loss? Do you mean that these spikes in SGD are caused by a large step size rather than the stochasticity of sampling? If yes, then how are spikes in the top eigenspace of the NTK an *evidence* spikes in SGD? The NTK is also influenced by the iterates, and the iterates are influenced by both the step size and the sampling. How do you disentangle the two effects?
> >
> > About Geiping et al., 2022:
> >
> > My main concern there is that the authors claimed "small batch leads to generalization" is one of the phenomenon being connected in this work, but since Geiping et al., 2022 shows that it's not always the case, the authors should make a comment about it in the paper.
> >
> > Step size across batches:
> > > Note that a larger batch size will make the gradient closer to the expectation, and a smaller batch size can have a higher variance in the gradient. To ensure the magnitude of the norm of weight update is similar, we use the same learning rate for different batches.
> >
> > I don't see why you need to ensure the norm of the weight update is similar. From an optimization perspective, shouldn't the step size be smaller when using a smaller batch, precisely to take care of the higher variance?

---

> > > ### Author Response · Authors · 2023-11-20
> > >
> > > *Comment 5: ``catapults show up in the top eigenspace of the NTK serves as evidence that the loss spikes in SGD are catapults``
> > > I still find this quite vague --- aren't catapults just spikes in the training loss, i.e., non-monotonically decreasing training loss? Do you mean that these spikes in SGD are caused by a large step size rather than the stochasticity of sampling? If yes, then how are spikes in the top eigenspace of the NTK an evidence spikes in SGD? The NTK is also influenced by the iterates, and the iterates are influenced by both the step size and the sampling. How do you disentangle the two effects?*
> > >
> > > To the best of our knowledge, we are the first to connect the spikes of training loss using SGD to catapults. We provided evidence that these spikes are catapults by showing the similar behaviors of NTK and loss of SGD with those of GD in the original catapult paper [Lewkowycz et al. 2020].
> > >
> > > To cause loss spikes in SGD, a large step size is necessary: otherwise, almost no spikes can be observed (for example see Figure 4(b) with $\eta = 0.1$). Without subsampling, it reduces to GD where a large step size can still cause catapult. With subsampling, the critical learning rates for each batch are generally different and the catapult can happen if the learning rate is larger than the critical one (see Figure 4(b) with $\eta = 0.8$).
> > >
> > > In SGD, similar to GD, we consider the NTK and training loss corresponding to the whole dataset. We observed that the top eigenvalue of the NTK  decreases when there is a spike in the training loss and the loss spikes are in the top eigenspace of the tangent kernel, similar to what we have observed in GD. Given the evidence, we show the loss spikes of SGD are catapults.
> > >
> > > *Comment 6: About Geiping et al., 2022: My main concern there is that the authors claimed "small batch leads to generalization" is one of the phenomenon being connected in this work, but since Geiping et al., 2022 shows that it's not always the case, the authors should make a comment about it in the paper.*
> > >
> > > We had added some discussion and a reference in the related work.
> > >
> > > *Comment 7: Step size across batches:
> > > Note that a larger batch size will make the gradient closer to the expectation, and a smaller batch size can have a higher variance in the gradient. To ensure the magnitude of the norm of weight update is similar, we use the same learning rate for different batches.
> > > I don't see why you need to ensure the norm of the weight update is similar. From an optimization perspective, shouldn't the step size be smaller when using a smaller batch, precisely to take care of the higher variance?*
> > >
> > > In the over-parameterized case, it is not necessary to decrease the step size for smaller bath sizes. See for example [1].
> > >
> > > 1. C. Liu et al. "Aiming towards the minimizers: fast convergence of SGD for over-parametrized problems." (NeurIPS 2023) arXiv:2306.02601 (2023)

---

> > > > ### Comment · Reviewer_QEoS · 2023-11-22
> > > >
> > > > > To cause loss spikes in SGD, a large step size is necessary:
> > > >
> > > > Shouldn't it be large relative to the variance, and so spikes in SGD is always dependent on the stochasticity as well?
> > > >
> > > > > In the over-parameterized case, it is not necessary to decrease the step size for smaller bath sizes. See for example [1].
> > > >
> > > > Thanks for pointing out this reference. It seems like this paper studies the interpolation regime, which often occurs with an over-parameterized model. Is it guaranteed that the models used in the current work can also interpolate the data its given? If not, would this still be a valid argument to use the same step size across batches in the case of SGD?

---

> ### Author Response · Authors · 2023-11-19
>
> *Concern 5.a: I don't see how catapult in the eigenspace explains the quick loss drop.*
>
> As a simple clarifying example, consider linear regression. The convergence speed of GD is determined by the condition number of the sample matrix $(XX^T)$ since the learning rate is determined by its largest eigenvalue. That is to say, the convergence speed is fast along the top eigendirections but is slow along the remaining directions. The neural networks are close to their linear approximations locally by Taylor's theorem with the tangent kernel as the sample matrix.  Therefore to result in such a quick loss drop, the change of the loss must be in the top eigendirections of the tangent kernel.  In contrast, if the catapults also occurred in the non-top eigendirections, i.e., the loss corresponding to the non-top eigendirections was large at the peak, since the learning rate is determined by the top eigenvalue, the convergence speed would be slow for non-top eigendirections hence the loss would decrease slowly.
>
> *Concern 5.b: I find this entire sentence  "This explains why the loss drops quickly to pre-spike levels from the peak of the spike" vague and handwavy. It's unclear what the authors mean by pre-spike levels. Is it the loss level at the iteration right before the spike occurs, or 50 iterations before the spike occurs? What drop rate is considered quick? Quick compared to what?*
>
> This is an informal qualitative description. We observed that the loss is much smaller at the iteration right before the spike occurs. Quick means in just a few iterations (typically less than 5). The decrease of the loss at the peak of the spike is much quicker than the decrease from the loss that is not in the spikes.
>
> *Concern 6.a: what exactly is NTK parameterization is unclear...I recommend that the authors explicitly describe what the model architecture is, what the initializations are (crucial to the NTK parameterization), etc...*
>
> In NTK parameterization, each weight parameter is i.i.d. drawn from the standard normal distribution, instead of having width-dependent variance. There will be a scaling factor $1/\sqrt{m}$ to control the scale of each neuron, where $m$ is the width. NTK parameterization is commonly used to analyze over-parameterized neural networks in literature, e.g.,[1,2,3].
>
>  We described the NTK parameterization in preliminaries, which follows from the original definition in [Jacot et al. 2018]. All the models considered in our work are standard architecutres hence should be easy to find corresponding information. We have added more information in the appendix.
>
> 1. S. Du S., et al. "Gradient Descent Provably Optimizes Over-parameterized Neural Networks." International Conference on Learning Representations. 2018.
> 2. Z. Ji, and T. Matus. "Polylogarithmic width suffices for gradient descent to achieve arbitrarily small test error with shallow ReLU networks." International Conference on Learning Representations. 2019.
> 3. J. Lee, et al. "Wide neural networks of any depth evolve as linear models under gradient descent." Advances in neural information processing systems 32 (2019).
>
> *Concern 6.b: Why is the NTK parameterization used? Is it crucial that you must use this to produce the results?*
>
> The NTK parameterization is not crucial for our results, and similar results can be produced using the Pytorch default parameterization (see Figure 12, 18, and 20 in the appendix).  We follow the parameterization used in [Lewkowycz et al. 2020], under which setting the training dynamics for wide neural networks is approximately linear with a small learning rate and catapults occur with a large learning rate. With NTK parameterization, each entry of the NTK does not scale with the width otherwise it becomes infinity for infinite width.
>
> *Concern 6.c: how did you choose the the step size used for Figure 3?*
>
> The exact numbers of the step size can be found in Appendix E. The learning rate is increased to be roughly $2\eta_{crit}$ (it can vary for different tasks) to generate extra catapults.
>
> Concern 7: The related work section needs more work, in the sense that it currently presents a laundry list of related works, but how they are related to each other and to the current work is inadequately discussed....at the end of the Catapult phase paragraph, instead of explaining what catapult is, which was already done in the intro...
>
>
> We have updated the draft to further clarify the contributions of related works. Due to its relevance for our work, we wanted to give a detailed definition of the catapult phenomenon and point out the difference and the relation between the catapult phase phenomenon and the kernel regime.

---

> > ### Comment · Reviewer_QEoS · 2023-11-19
> >
> > > The convergence speed of GD is determined by the condition number of the sample matrix $XX^T$
> >
> > Do you mean $X^TX$? And even for linear regression, the Hessian and the NTK are different (they are not even of the same size). How are the eigenspaces comparable?
> >
> > > This is an informal qualitative description. We observed that the loss is much smaller at the iteration right before the spike occurs. Quick means in just a few iterations (typically less than 5). The decrease of the loss at the peak of the spike is much quicker than the decrease from the loss that is not in the spikes.
> >
> > It'd be helpful to make this more formal or at least include this in the paper (along with the NTK parameterization descriptions).
> >
> > >  under which setting the training dynamics for wide neural networks is approximately linear with a small learning rate and catapults occur with a large learning rate.
> >
> > I think I've already commented on this earlier, but I still don't understand why you used the NTK parameterization other than someone else also used it, and why does it even matter if model is approximately linear under NTK with a small learning rate, when you're in fact using a large learning rate for catapult to occur.
> >
> > > The learning rate is increased to be roughly $2/\eta_{crit}$
> >
> > So how do you increase it to be *roughly* that value? Could you point me to where it's described in the appendix? (There's quite a lot of stuff scattered around in the appendix and I don't think I was able to find it). This is very important for reproducibility.

---

> ### Author Response · Authors · 2023-11-19
>
> *Q1: Why does the critical step size only depend on the initialization and not varying throughout training?*
>
> The critical step size can definitely vary through training, which is exactly what we leveraged to induce multiple catapults in GD (see Figure 3).  We can observe multiple catapults with the same step size in SGD due to fluctuations in the critical learning rate due to the stochasticity of the batch.
>
> *Q2: Paragraph under Claim 1: shouldn't it be $\mathcal{PL}_n$ instead of $\mathcal{PL}_5$ that corresponds to the spike in the training loss? Is even $\mathcal{PL}_5$ plotted somewhere that I missed?*
>
> In Figure 5, we plotted the loss corresponding to the complement of the top-5 eigendirections of the tangent kernel, i.e., $\mathcal{PL}_5^\bot$ which decreases almost monotonically. Therefore, $\mathcal{PL}_5$, which is $\mathcal{L}-\mathcal{PL}_5^\bot$ corresponds to the spikes in the training loss.
>
> *Q3.a: Choice of top eigenspace dimension $s$: The first sentence after this - I'm confused by what this sentence is trying to convey, especially the second part. What linear dynamics?*
>
> For a clarifying example, consider a linear model $h(w;x) = w^Tx$ where $w$ is the parameter. For $n$ data points $(X,y)$, by minimizing the squared loss $L(w) = \frac{1}{n}||{h(w;X) - y}||^2$ its training dynamics of GD with learning rate $\eta$ satisfies the equation:
> \begin{align*}
>   h(w_{t+1};X) - y = (I - \eta \frac{2}{n}XX^T) (h(w_t;X) - y).
> \end{align*}
> Then we project the residual $h(X)-y$ to the eigenspace of $XX^T$, which we denote by $u_i\in\mathbb{R}^n$ and we get:
> \begin{align*}
>     u_i^T( h(w_{t+1};X) - y) = (1 - \eta\frac{2\lambda_i}{n})u_i^T( h(w_{t};X) - y),
> \end{align*}
> where $\lambda_i$ is the i-th eigenvalue of $XX^T$.
>
> Note that if $\eta > n/\lambda_i$, the loss will increase along the eigen-direction $u_i$.
>
> As wide neural networks are close to their linear approximations before the spike, they follow approximately linear dynamics. The sample matrix in this case is the tangent kernel. Then a larger learning rate will lead to the increase of the loss in more eigen-directions.
>
> *Q3.b:  Do you have an explanation to why small $s$ contradicts to what you expect?*
>
> We find this observation interesting since the dramatic change of the loss turns out to occur in a low-dimensional subspace.  Moreover, this phenomenon is quite consistent among our experiments covering a broad class of neural networks.
>
> *Q3.c:  Is there a way to verify $s$ corresponds to support vectors?*
>
> Could you please clarify what you mean by the connection between $s$ and ''support vectors''? We are not sure we understood your comment about the ''support vectors''.
>
> *Q4: $f(X_{batch})$ has the same length as the number of examples in the batch. But the projection operator acts on prediction vectors of length $n$.*
>
> We further clarify this point.  The projection operator is defined based on the tangent kernel corresponding to the mini-batch, hence it acts on the prediction vectors of length $b$. We have made it clear in the revision. Specifically, we edited the beginning of section 3.1: we define the network output $f$ first then define the corresponding NTK and the training loss. In this way, the projection operator depends on the length of the output.
>
> *Q5: Do you need to recompute $\eta_{crit}(X_{batch})$ for each batch? How does it work?*
>
> We recompute $\eta_{crit}(X_{batch})$ for each batch. For a mini-batch $X_{batch}$, we can compute the batch loss $L(X_{batch})$ and the corresponding Hessian of the loss $H_L(X_{batch})$. The critical learning rate is computed as $2/||H_L(X_{batch})||_2$.
>
> *Q6: Does high AGOP alignment imply high generalization in low-rank matrix sensing problem?*
>
> This is a very interesting example for understanding AGOP alignment.  In fact, this is a problem we are currently working on but the results are out of scope of this submission.
>
> We thank you for the minor suggestions. We have revised our draft according to your suggestions.
>
> *Minor suggestions: ...To me it's more like Cohen et al. "observed" numerous catapults at EoS, but conjectured that EoS and catapults have the same underlying cause.*
>
> Indeed in [Cohen et al. 2020] they observed numerous loss spikes but they did not provide any evidence that these are catapults. We have made it clear in our revised draft.
>
> *Minor suggestions: When defining the AGOP alignment in page 4, the "true model" $f$ should be defined more rigorously...*
>
> We actually made it clear that when the true model is not available, we use a SOTA model as a substitute in footnote 1 at the end of page 2. We will mention it again when we define true AGOP to avoid confusion.

---

> > ### Comment · Reviewer_QEoS · 2023-11-19
> >
> > Paragraph under Claim 1: Thanks for the explanation. I think I was caught up in the wording: it's not the spikes in the training loss, rather the spikes in the training loss in the top 5 eigenspace of K. (I'm assuming you meant Figure 2, not 5). Also I just noticed a typo in Figure 5' s caption: cataput -> catapult
> >
> > Please see my comment above for an explanation of what I mean by "support vectors".

---

> > > ### Author Response · Authors · 2023-11-20
> > >
> > > *Comment 12: Paragraph under Claim 1: Thanks for the explanation. I think I was caught up in the wording: it's not the spikes in the training loss, rather the spikes in the training loss in the top 5 eigenspace of K. (I'm assuming you meant Figure 2, not 5). Also I just noticed a typo in Figure 5' s caption: cataput -> catapult*
> > >
> > > That is correct, thanks for noticing the typo, we will clarify.

---

> ### Comment · Reviewer_QEoS · 2023-11-19
>
> (Sorry, I don't know why it's not showing as part of a thread...)
>
> Thank you for the reply. The addition of Figure 15 is nice, but since it's ran with SGD, it's unclear to me whether the catapults are due to stochasticity in the sampling or due to having a large step size. How was the step size selected here? And does more catapults also lead to higher accuracy? From this plot I can only tell that for the logistic loss, there's also a correlation between catapults and drops in $\|K\|_2$. Does it also correlate with higher accuracy?
>
> What I mean by support vectors is that - since you are doing classification with the squared loss under some neural network, the model represents some nonlinear separator of the data into classes. The examples that are closest to this separator can be thought of as the support vectors, just like in the linear case (SVM). So if we take a large step such that a catapult is to occur for the training loss, then it ought to be these "difficult examples" that are affected the most. No each entry in the NTK measures the similarity between two examples when it comes to how much their prediction changes by a change in the model. And so it's natural for me to think that when a catapult occurs, the loss along the eigendirections corresponding to these "support vectors" will also catapult, hence why I find it somewhat unsurprising. Regarding your comment:
> > as the catapult dynamics of complicated neural networks, including Wide ResNets and Vision Transformer, indeed occur in a low dimensional subspace
>
> The NTK matrix is $n\times n$ where $n$ is the number of training examples. How exactly is a subspace spanned by the eigenvectors of the NTK related to the dimension of the neural network? I think it's more related to a subset of the examples.
>
> I agree that the NTK is closely related to the Hessian of the loss. Can you explain why
> > ...shares the same eigenvalue with NTK with extra ...
>
> What if $n>p$? And why is it ok to assume $f_i-y_i=O(1)$, especially when you are studying catapults where this quantity is large? Please correct me if I'm wrong, but afaik the kernel regime is when the step size is tiny so that the model doesn't move much and hence the linear approximation is accurate. Doesn't it contradict to what it requires for catapults to occur?

---

> > ### Author Response · Authors · 2023-11-20
> >
> > We thank you for the rely and insightful comments. We will address your concerns below.
> >
> > *Comment 1: ...since it's ran with SGD, it's unclear to me whether the catapults are due to stochasticity in the sampling or due to having a large step size. How was the step size selected here? And does more catapults also lead to higher accuracy? From this plot I can only tell that for the logistic loss, there's also a correlation between catapults and drops in $||K||_2$. Does it also correlate with higher accuracy?*
> >
> > The step size in the plot was chosen the same as the experiment for the squared loss.
> >
> > We don't make any claims about catapults with logistic loss in this paper. While these plots are suggestive, as we said, further evidence is needed to show that the loss spikes in the logistic loss are indeed catapults.  The full exploration of this issue is beyond the scope of our work.
> >
> > *Comment 2: What I mean by support vectors is that - since you are doing classification with the squared loss under some neural network, the model represents some nonlinear separator of the data into classes. The examples that are closest to this separator can be thought of as the support vectors, just like in the linear case (SVM). So if we take a large step such that a catapult is to occur for the training loss, then it ought to be these "difficult examples" that are affected the most. No each entry in the NTK measures the similarity between two examples when it comes to how much their prediction changes by a change in the model. And so it's natural for me to think that when a catapult occurs, the loss along the eigendirections corresponding to these "support vectors" will also catapult, hence why I find it somewhat unsurprising.*
> >
> > Thanks for your clarification. We believe the "support vectors" you mentioned is different from the top eigen-diretions of the tangent kernel discussed in our paper since they are in different spaces: the "support vectors" are in the feature space, i.e., the space spanned by the non-linear transformation on the input data, while the top eigenvectors are in the eigenspace of the tangent kernel. Therefore, we are not sure if our findings on the eigenspace of the tangent kernel can be observed in the feature space as well.  Furthermore, while for the hinge loss, support vectors are usually a fraction of the data, in the case of the square loss all coefficients are generally non-zero. Thus identifying ``support vectors'' requires an additional thresholding parameter.
> >
> > *Comment 3: Regarding your comment:
> > ``as the catapult dynamics of complicated neural networks, including Wide ResNets and Vision Transformer, indeed occur in a low dimensional subspace.`` The NTK matrix is $n\times n$ where  $n$ is the number of training examples. How exactly is a subspace spanned by the eigenvectors of the NTK related to the dimension of the neural network? I think it's more related to a subset of the examples.*
> >
> > By low dimensional subspace, we mean the space spanned by the top eigenvectors of the tangent kernel.
> >
> > *Comment 4:  Can you explain why
> > ``...shares the same eigenvalue with NTK with extra ...``
> > What if  $n>p$? And why is it ok to assume $f_i - y_i = O(1)$, especially when you are studying catapults where this quantity is large? Please correct me if I'm wrong, but afaik the kernel regime is when the step size is tiny so that the model doesn't move much and hence the linear approximation is accurate. Doesn't it contradict to what it requires for catapults to occur?*
> >
> > We considered over-parameterized neural networks in our paper hence $p>n$.
> >
> > We did not claim that $f_i - y_i = O(1)$ holds during catapult dynamics. It is not generally true when catapults occur since the loss can be large. However, during the catapult dynamics, we empirically observed that the critical learning rate estimated from NTK is close to the critical learning rate estimated from loss Hessian (please see Figure 9 in Appendix A for the experimental results). Furthermore, NTK is shown to be a useful quantity for analyzing catapult dynamics (see e.g., [1,2,3]).
> >
> > 1. A. Lewkowycz, et al. "The large learning rate phase of deep learning: the catapult mechanism." arXiv preprint arXiv:2003.02218 (2020).
> >
> > 2. L. Zhu, et al. "Quadratic models for understanding neural network dynamics." arXiv preprint arXiv:2205.11787 (2022).
> >
> > 3. D. Meltzer, and L. Junyu . "Catapult Dynamics and Phase Transitions in Quadratic Nets." arXiv preprint arXiv:2301.07737 (2023).

---

> > > ### Comment · Reviewer_QEoS · 2023-11-22
> > >
> > > Thanks for the clarifications.
> > >
> > > I understand that the support vectors are features, whereas the eigenvectors of the NTK can be in a different space, but what I was wondering is whether there is a correspondence between the two, as there are n data points and the NTK is an nxn matrix.
> > >
> > > Regarding the thread on
> > > > Comment 3: Regarding your comment: `as the catapult dynamics of complicated neural networks, including Wide ResNets and Vision Transformer, indeed occur in a low dimensional subspace.` The NTK matrix is where is the number of training examples. How exactly is a subspace spanned by the eigenvectors of the NTK related to the dimension of the neural network?I think it's more related to a subset of the examples.
> > > >> By low dimensional subspace, we mean the space spanned by the top eigenvectors of the tangent kernel.
> > >
> > > The NTK allows us to view the features of each example as a p-dimensional vector, where p is the size of the network. But the NTK can have at most n eigenvectors, and by overparameterization we already have n << p. I don't understand how the two spaces are comparable?
> > >
> > > Last comment about overparamterization: I understand that you do not claim $f_i-y_i=O(1)$, but that is what you used earlier to argue that the NTK and the Hessian are closely related, so perhaps that is not a valid argument.

---

> ### Author Response · Authors · 2023-11-20
>
> *Comment 8: ``The convergence speed of GD is determined by the condition number of the sample matrix $XX^T$.``
> Do you mean $X^T X$? And even for linear regression, the Hessian and the NTK are different (they are not even of the same size). How are the eigenspaces comparable?*
>
> For a linear model $h(w;x) = w^Tx$ where $w$ is the parameter. For $n$ data points $(X,y)$ where $X\in \mathbb{R}^{n\times d}$, $y\in\mathbb{R}^n$, consider minimizing the squared loss $L(w) = \frac{1}{n}||{h(w;X) - y}||^2 = \frac{1}{n}|| Xw - y||^2$.
>
> The loss Hessian takes the form $H_L = \nabla^2 L(w) = X^T X \in \mathbb{R}^{d\times d}$, and the NTK takes the form $K = \frac{d h(w;X)}{dw} \frac{d h(w;X)}{dw}^T = XX^T \in \mathbb{R}^{n\times n}$.
>
> $XX^T$ and $X^TX$ share the same non-zero eigenvalues.
>
> *Comment 9: ``This is an informal qualitative description. We observed that the loss is much smaller at the iteration right before the spike occurs. Quick means in just a few iterations (typically less than 5). The decrease of the loss at the peak of the spike is much quicker than the decrease from the loss that is not in the spikes.``
> It'd be helpful to make this more formal or at least include this in the paper (along with the NTK parameterization descriptions).*
>
> We have updated the draft to add a description of the quick loss drop in the introduction and NTK parameterization in Appendix F.
>
> *Comment 10: ``under which setting the training dynamics for wide neural networks is approximately linear with a small learning rate and catapults occur with a large learning rate.``
> I think I've already commented on this earlier, but I still don't understand why you used the NTK parameterization other than someone else also used it, and why does it even matter if model is approximately linear under NTK with a small learning rate, when you're in fact using a large learning rate for catapult to occur.*
>
> We also used Pytorch default initialization in our work (see Figure 12, 18, and 20 in the appendix). The original catapult paper [1], as well as the analysis of the catapults on simplified models (e.g., [2,3]) suggest that catapults occur using NTK parameterization with a large learning rate.    Our work used this well-established setting to study the impact of the catapults.
>
> Furthermore, the analyses in [1,2,3] also suggest that models that are approximately linear with a small learning rate (including Pytorch default initialization [4]) can have catapults with a large learning rate. Therefore the kernel regime is closely related to the catapults.
>
> Additionally, we note that small and large learning rates are generally only different by a factor of two (whether it is smaller or larger than the critical learning rate).
>
> 1. A. Lewkowycz, et al. "The large learning rate phase of deep learning: the catapult mechanism." arXiv preprint arXiv:2003.02218 (2020).
>
> 2. L. Zhu, et al. "Quadratic models for understanding neural network dynamics." arXiv preprint arXiv:2205.11787 (2022).
>
> 3. D. Meltzer, and L. Junyu . "Catapult Dynamics and Phase Transitions in Quadratic Nets." arXiv preprint arXiv:2301.07737 (2023).
>
> 4. G. Yang,  and H. Edward J.. "Feature learning in infinite-width neural networks." arXiv preprint arXiv:2011.14522 (2020).
>
> *Comment 11: ``The learning rate is increased to be roughly $2/\eta_{crit}$``.
> So how do you increase it to be roughly that value? Could you point me to where it's described in the appendix? (There's quite a lot of stuff scattered around in the appendix and I don't think I was able to find it). This is very important for reproducibility.*
>
> The learning rates and the iterations are specified in Appendix F( Experimental details), in the paragraphs corresponding to Figure 3 and Figure 6.

---

> > ### Comment · Reviewer_QEoS · 2023-11-22
> >
> > Thanks for the clarification, what I meant was $X^TX$ is the Hessian rather than $XX^T$, and I agree that the eigenvalues are the same up to 0's and the eigen basis corresponding to non-zero eigenvalues are the same up to a transformation. However, I still find the entire argument inconsistent: in my original review, I was wondering what you mean by linear dynamics in the sentence
> > > Note that a larger learning rate will make the loss increase in more eigen-directions, as indicated by the linear dynamics before the spike.
> >
> > to which you replied
> > > As wide neural networks are close to their linear approximations before the spike, they follow approximately linear dynamics.
> >
> > I think I've raised this before as well --- you need the learning rate to be small to "follow approximately linear dynamics", but the basis of having catapults is to use s large learning rate.

---

> > ### Comment · Reviewer_QEoS · 2023-11-22
> >
> > Thanks for pointing out where you described how the learning rate is increased.
> >
> > I strongly recommend reorganizing the entire paper in terms of where figures are placed, and where the experiment details for each figure are. At its current state, it's extremely difficult to find the experiment setup for a particular experiment without jumping three times across two pdf files, as the descriptions all cross reference each other. Some grouping for the experiment details can be helpful, and repeated information may seem redundant but can be helpful in quickly finding all the information one needs for a particular experiment.

---

> > > ### Author Response · Authors · 2023-11-22
> > >
> > > We thank you for your rely. We will address your concerns below.
> > >
> > > *Comment 1:  I understand that the support vectors are features, whereas the eigenvectors of the NTK can be in a different space, but what I was wondering is whether there is a correspondence between the two, as there are n data points and the NTK is an nxn matrix.*
> > >
> > > This is an interesting question but we are not aware of any direct correspondence.
> > >
> > > *Comment 2: Regarding the thread on\\
> > > ``Comment 3: Regarding your comment: as the catapult dynamics of complicated neural networks, including Wide ResNets and Vision Transformer, indeed occur in a low dimensional subspace. The NTK matrix is where is the number of training examples. How exactly is a subspace spanned by the eigenvectors of the NTK related to the dimension of the neural network?I think it's more related to a subset of the examples.``,\\
> > > ``By low dimensional subspace, we mean the space spanned by the top eigenvectors of the tangent kernel.
> > > ``\\The NTK allows us to view the features of each example as a p-dimensional vector, where p is the size of the network. But the NTK can have at most n eigenvectors, and by overparameterization we already have n << p. I don't understand how the two spaces are comparable?*
> > >
> > > We do not consider feature space, i.e., $\mathbb{R}^p$ but consider the tangent kernels $\mathbb{R}^{n\times n}$ corresponding to these networks. The space spanned by the top eigenvectors of the tangent kernel is a subspace of $\mathbb{R}^n$ where $n$ is the sample size.
> > >
> > > *Comment 3: Last comment about overparamterization: I understand that you do not claim $f_i - y_i =O(1)$, but that is what you used earlier to argue that the NTK and the Hessian are closely related, so perhaps that is not a valid argument.*
> > >
> > > The argument is primarily empirical. In the literature, it has been shown that NTK is a useful quantity to analyze catapult dynamics[1,2,3].   In Appendix A, we empirically show that when training with catapult dynamics,  the top eigenvalues of NTK divided by $n$ and the top eigenvalue of the loss Hessian are close, which can be seen from the estimation of critical learning rates based on these two quantities. However, the theoretical argument shows that in the NTK regime, these two quantities are indeed closely related[4].
> > >
> > > 1. A. Lewkowycz, et al. "The large learning rate phase of deep learning: the catapult mechanism." arXiv preprint arXiv:2003.02218 (2020).
> > >
> > > 2. L. Zhu, et al. "Quadratic models for understanding neural network dynamics." arXiv preprint arXiv:2205.11787 (2022).
> > >
> > > 3. D. Meltzer, and L. Junyu . "Catapult Dynamics and Phase Transitions in Quadratic Nets." arXiv preprint arXiv:2301.07737 (2023).
> > >
> > > 4.  C. Liu, et al. "On the linearity of large non-linear models: when and why the tangent kernel is constant." Advances in Neural Information Processing Systems 33 (2020): 15954-15964.

---

> > > > ### Author Response · Authors · 2023-11-22
> > > >
> > > > *Comment 4: ``To cause loss spikes in SGD, a large step size is necessary:``\\
> > > > Shouldn't it be large relative to the variance, and so spikes in SGD is always dependent on the stochasticity as well?*
> > > >
> > > > If there is no variance, i.e., the critical learning rates are the same for each batch,  a large learning rate will still cause catapult dynamics. This follows directly from the mechanism of catapults in GD.
> > > >
> > > > Larger variance in smaller batch sizes allows more critical learning rates to be smaller than the learning rate, hence inducing more catapults.
> > > >
> > > > *Comment 5: ``In the over-parameterized case, it is not necessary to decrease the step size for smaller bath sizes. See for example [1].``\\ Thanks for pointing out this reference. It seems like this paper studies the interpolation regime, which often occurs with an over-parameterized model. Is it guaranteed that the models used in the current work can also interpolate the data its given? If not, would this still be a valid argument to use the same step size across batches in the case of SGD?*
> > > >
> > > > The models considered in our work can interpolate the data. We trained all  models  to reach near $0$ training loss. For example, Figure 22 shows the training loss corresponding to Rank-2 and Rank-4 regression with batch size $5$, where the training loss reaches $0$. We will add the plots for other tasks in the revision.

---

> > > > > ### Author Response · Authors · 2023-11-22
> > > > >
> > > > > *Comment 6: Thanks for the clarification, what I meant was $X^T X$ is the Hessian rather than $XX^T$, and I agree that the eigenvalues are the same up to 0's and the eigen basis corresponding to non-zero eigenvalues are the same up to a transformation. However, I still find the entire argument inconsistent: in my original review, I was wondering what you mean by linear dynamics in the sentence \\``Note that a larger learning rate will make the loss increase in more eigen-directions, as indicated by the linear dynamics before the spike.``\\ to which you replied\\ ``As wide neural networks are close to their linear approximations before the spike, they follow approximately linear dynamics.``\\ I think I've raised this before as well --- you need the learning rate to be small to "follow approximately linear dynamics", but the basis of having catapults is to use s large learning rate.*
> > > > >
> > > > > For wide neural networks linear dynamics holds for any learning rate smaller than the critical learning rate not just a small learning rate.
> > > > >
> > > > > For eigendirections whose eigenvalues are larger than $2/\eta$, there will be catapults; for eigendirections whose eigenvalues are smaller than $2/\eta$, there will be no catapult. To the best of our knowledge, this has not been observed in the literature.

---

> > > > > > ### Author Response · Authors · 2023-11-22
> > > > > >
> > > > > > *Comment 7: Thanks for pointing out where you described how the learning rate is increased.
> > > > > > I strongly recommend reorganizing the entire paper in terms of where figures are placed, and where the experiment details for each figure are. At its current state, it's extremely difficult to find the experiment setup for a particular experiment without jumping three times across two pdf files, as the descriptions all cross reference each other. Some grouping for the experiment details can be helpful, and repeated information may seem redundant but can be helpful in quickly finding all the information one needs for a particular experiment.*
> > > > > >
> > > > > > Thank you for the suggestion.
> > > > > > We have grouped the experimental details based on the position of the figures in the main text and edited the caption of the figures accordingly.

---

### Official Review · Reviewer_iMUj · 2023-10-30

**Soundness:** 3 good
**Presentation:** 4 excellent
**Contribution:** 3 good
**Rating:** 8
**Confidence:** 4

**Summary:**

The paper analyses the spikes in the learning curve of SGD in DNNs. They provide empirical evidence that:
- These spikes in the loss correspond to an increase in the error along the top eigenvalues of the NTK.
- The spikes happen when the learning rate becomes larger than the critical learning rate (which can depend on the batch).
- The number of spikes (which depends on the learning rate, batch-size and training algorithm) is well corelated with the test error and the AGOP alignement.

**Strengths:**

We are still lacking a good understanding of why these spikes appear during training, and even more why they improve performences. The paper proposes three simple (but to my knowledge new) postulates and checks them on a variety of tasks and architectures. These ideas form together a pretty complete picture.

**Weaknesses:**

The paper provides evidence that the spikes help generalization, but not so much in terms of what happens after the spikes that improve generalization.

For example, though the paper shows that the NTK decreases in size after each spike, it does not describe how it is changing, one could expect the NTK to decrease more along its top eigenvalues for example, since these eigenvalues are highly related to the spikes.

The only explanation that is proposed is that the AGOP alignement increases, but it is not explained why these spikes should increase AGOP alignement. Also while the paper presents the AGOP alignement as `the mechanism throught which NN learn features', I personnaly think that while the AGOP is an interesting measure of feature learning, but it is unlikely that it fully captures feature learning.

The authors could have considered other measures of feature learning, such as the alignement between the NTK and the task (there are multiple possible alignement measures). This could have been especially interesting since the first part of the argument suggest that the NTK spectrum plays an important role in the spikes.

**Questions:**

I would prefer if you did not present the AGOP as `the mechanism throught which NN learn features'. I think that there is clearly not enough evidence today to make this claim, when multiple other mechanisms for feature learning have been proposed (for example Information Bottleneck, NTK/kernel alignement). While the AGOP alignement seems well suited to low-index tasks (since the `index' can be recovered from the AGOP), it seems ill suited to measure feature learning when the learned features depend nonlinearly on the inputs, such as in (https://arxiv.org/abs/2305.19008).

---

> ### Author Response · Authors · 2023-11-18
>
> We thank the reviewer for the positive feedback and insightful comments. We will address your concerns and questions below.
>
> *W1:   it does not describe how NTK is changing...one could expect the NTK to decrease more along its top eigenvalues for example, since these eigenvalues are highly related to the spikes.*
>
> This is a nice point and we believe this is indeed correct. We conducted an experiment by training a two-layer neural network on 128 CIFAR10 data points. The change in the eigenvalues of NTK is as follows (we report the average of 3 independent runs):
>
> |     Index of eigenvalue of NTK  |      1    | 2   | 3   | 4   | 5  |
> |-------|-----------|-----------|----------|----------|----------|
> | Change (%) | -55.2      | -37.2      | -36.0     | -37.21    | -22.67   |
>
> The reasons we only show the change of the spectral norm (i.e., the top eigenvalue) of the tangent kernel are  the following:
>
> 1) the catapult phase is characterized by the decrease of the spectral norm of the tangent kernel in the original paper  [Lewkowycz et al. 2020] for GD. We use this characterization to show that catapults occur in SGD as well.
>
> 2) the top eigenvalue relates directly to the optimization properties since the critical learning rate is determined by the top eigenvalue.
>
> *W2: The only explanation that is proposed is that the AGOP alignment increases, but it is not explained why these spikes should increase AGOP alignment.*
>
> The mechanism of how these spikes increase AGOP alignment can be seen as follows.  In the literature, e.g., [3], the change of the NTK is closely related to feature learning. Note that with $0$ catapult, i.e., NTK regime, due to the small weight change, AGOP is not able to update a lot to align with the true AGOP.  In work[1,2], it was proved that NTK models cannot effectively learn features.  However, with catapults, the weights are allowed to change a lot (as the gradient of the loss i.e., $dL/dw = (f-y)\frac{df}{dw}$ scales with the value of the loss, which is huge when catapults occur). Consequently, it leads to a significant change in the AGOP. Please see our experiments of multiple catapults in GD (Figure 6), where a greater number of catapults in GD leads to a higher (better) AGOP alignment and smaller (better) test loss/error. This mechanism is similar to the feature learning process with near-zero initialization[3], in which case the AGOP is also allowed to change a lot during training  (validated by our experimental results in Figure 16). Note that to get a better generalization/AGOP alignment, it is necessary to have a significant AGOP change when the true AGOP is low-rank (e.g., multi-index models), compared to the full-rank AGOP at initialization.  And this can be realized by having catapults.  Our experiments on real-world datasets such as SVHN and CelebA suggest that this is the case of how neural networks learn features in practice. A more detailed explanation is in the paragraph " Intuition of the connection between catapults and AGOP alignment" after Figure 6.
>
> 1. B. Ghorbani, et al. "Limitations of lazy training of two-layers neural network." Advances in Neural Information Processing Systems 32 (2019).
> 2. C. Wei,  et al. "Regularization matters: Generalization and optimization of neural nets vs their induced kernel." Advances in Neural Information Processing Systems 32 (2019).
> 3. G. Yang,  and H. Edward J.. "Feature learning in infinite-width neural networks." arXiv preprint arXiv:2011.14522 (2020).
>
> *W3: ...it is unlikely that AGOP alignment fully captures feature learning. The authors could have considered other measures of feature learning, such as the alignment between the NTK and the task. This could have been especially interesting ...*
>
> *Q1: I would prefer if you did not present the AGOP as the mechanism throught which NN learn features...  While the AGOP alignement seems well suited to low-index tasks, it seems ill suited to measure feature learning when the learned features depend nonlinearly on the inputs, such as in (https://arxiv.org/abs/2305.19008).*
>
> Thank you for the suggestion and the reference.  We note that AGOP has been shown to accurately capture features learned in a broad class of neural network architectures, including both nonlinear fully connected and convolutional networks (see [1, 2]).  Given this line of work and our results in this paper, we believe that a major contribution to neural feature learning comes from AGOP.  Nevertheless, we feel connecting AGOP alignment and other measures of generalization is an important direction of future work.  We have added a discussion of the suggested references to our draft.
>
> 1. A. Radhakrishnan, et al. "Feature learning in neural networks and kernel machines that recursively learn features." arXiv preprint arXiv:2212.13881 (2022).
>
> 2. D. Beaglehole, et al. "Mechanism of feature learning in convolutional neural networks." arXiv preprint arXiv:2309.00570 (2023).

---

### Official Review · Reviewer_LhuJ · 2023-10-31

**Soundness:** 3 good
**Presentation:** 3 good
**Contribution:** 3 good
**Rating:** 6
**Confidence:** 4

**Summary:**

This paper discusses spikes in training loss dynamics when neural networks are trained with SGD, attributing them to "catapults" - an optimization phenomenon noted in GD with large learning rates in one previous work. They demonstrate that these catapults exist in a subspace defined by top eigenvectors of the tangent kernel for both GD and SGD. They also suggest that catapults enhance generalization by promoting feature learning and alignment with the AGOP of the true predictor. Additionally, they show that a smaller SGD batch size results in more catapults, thus boosting AGOP alignment and test results.

**Strengths:**

It's important to know why there are spikes in the training loss of modern NNs. This paper offers a clear and organized explanation. I think this will be very helpful for the community. The article is well-written and structured.

**Weaknesses:**

This paper lacks proof. It proposes a potential reason for spikes in neural network training loss but doesn't provide any rigorous theoretical analysis. Can we do this theory in a simpler setting? Furthermore, the authors seem to assume a linearized dynamics between spikes, but I'm not sure if this holds true in real-world situations. I don't think this is true in real-word situations, and I don't know whether this is true in simpler cases. For instance, in a basic setup using a 2-layer network, does this apply? While I understand the main idea (that with a higher learning rate, GD moves towards flatter regions to ensure the stability condition, which causes spikes, and with SGD's randomness, spikes might occur more often because the stability condition is changing), I'm not convinced about the linearized approximation between two spikes. Can you clarify this? If this is addressed, I'd rate the paper higher to 8.

**Questions:**

No.

---

> ### Author Response · Authors · 2023-11-18
>
> We thank the reviewer for the positive feedback and insightful comments. We will address your concerns and questions below.
>
> *W1: No rigorous theoretical analysis for spikes in neural network training loss.*
>
> Indeed we do not analyze the mechanism underlying the catapult phenomenon in this paper but rather focus on their implications. It is a well-established phenomenon first identified in  [Lewkowycz et al. 2020]. We refer the reviewer to, e.g., works [1,2,3] for theoretical analyses in simplified models.
>
> 1. A. Lewkowycz, et al. "The large learning rate phase of deep learning: the catapult mechanism." arXiv preprint arXiv:2003.02218 (2020).
>
> 2. L. Zhu, et al. "Quadratic models for understanding neural network dynamics." arXiv preprint arXiv:2205.11787 (2022).
>
> 3. D. Meltzer, and L. Junyu. "Catapult Dynamics and Phase Transitions in Quadratic Nets." arXiv preprint arXiv:2301.07737 (2023).
>
> *W2: Not sure if the linearized approximation between two spikes holds in real-world situations.*
>
> It looks like the reviewer's comment refers to our Remark 2, which does appear to be cryptic. We clarify that we are not assuming the linear dynamics hold between the spikes. We only used the fact that wide neural networks are locally linear in a small neighborhood to estimate the critical learning rate (please see Claim 2 where the critical learning rate depends on iteration $t$).   We have updated our draft to clarify this point.

---

### Official Review · Reviewer_Ykc6 · 2023-10-31

**Soundness:** 3 good
**Presentation:** 3 good
**Contribution:** 3 good
**Rating:** 6
**Confidence:** 4

**Summary:**

This paper provides an empirical verification of the claim that catapults in SGD occur in the top eigenspace of the NTK, where the catapults correspond to the "spikes" of the neural network training. Moreover, by leveraging the AGOP alignment, the paper also claims that more catapults induce better test accuracies. These claims are connected with some of the neural network training hyperparameters, most notably the batch size, and are corroborated with extensive experiments.

**Strengths:**

1. The paper is well-motivated and well-written
2. The experiments generally support the claims regarding catapults and generalization
3. The paper has some theoretical discussions.

**Weaknesses:**

1. The paper never answers *why* the catapults occur. The observations that the catapults happen in the top eigenspace of NTK and that it leads to a decrease in the spectral norm of NTK are not the actual mechanisms behind why catapults occur. Rather, they are direct conclusions of catapults occurring.
2. The authors discuss in detail that the number of catapults correlates strongly with AGOP, which is known to be correlated strongly with test accuracy. But to me, these are still just correlations not explicit inner workings of catapults and their effects on generalization.

**Questions:**

1. Any results on (stochastic) gradient descent with momentum?
2. Although the authors did provide sufficient motivation for considering AGOP, I'm still curious about the relations to other "generalization measures" such as flatness (max eigenvalue of Hessian), tail-index of the trajectory/weight matrix [2,3]...etc.
3. Any connection to recently uncovered saddle-to-saddle hopping dynamics [3,4]? Can the hoppings be equated to catapults?


[1] https://proceedings.neurips.cc/paper/2020/hash/37693cfc748049e45d87b8c7d8b9aacd-Abstract.html
[2] https://jmlr.org/papers/v22/20-410.html
[3] https://arxiv.org/abs/2304.00488
[4] https://arxiv.org/abs/2106.15933

---

> ### Author Response · Authors · 2023-11-18
>
> We thank the reviewer for the positive feedback and insightful comments. We will address your concerns and questions below.
>
> *W1: The paper never answers why the catapults occur...are not the actual mechanisms behind why catapults occur. Rather, they are direct conclusions of catapults occurring.*
>
> Indeed we do not analyze the mechanism underlying the catapult phenomenon in this paper but rather focus on their implications.  It is a well-established phenomenon first identified in  [Lewkowycz et al. 2020]. We refer the reviewer to, e.g., works [1,2,3] for theoretical analyses in simplified models.
>
> 1. A. Lewkowycz, et al. "The large learning rate phase of deep learning: the catapult mechanism." arXiv preprint arXiv:2003.02218 (2020).
>
> 2. L. Zhu, et al. "Quadratic models for understanding neural network dynamics." arXiv preprint arXiv:2205.11787 (2022).
>
> 3. D. Meltzer, and L. Junyu. "Catapult Dynamics and Phase Transitions in Quadratic Nets." arXiv preprint arXiv:2301.07737 (2023).
>
> *W2: The correlation between the number of catapults and AGOP is not explicit inner workings of catapults and their effects on generalization.*
>
> We believe there is a causal relation between catapults and generalization. This is verified in our experiments, particularly in  Fig. 6, where we induce catapults by increasing the learning rate and each consecutive catapult leads improved generalization.  We empirically show that after each catapult, AGOP alignment improves, and it is well-known that better AGOP alignment leads to improved generalization (see [1, 2, 3, 4, 5]).
>
> 1. W. Härdle, and S. Thomas M. "Investigating smooth multiple regression by the method
> of average derivatives." Journal of the American statistical Association 84.408 (1989): 986-995.
> 2. M. Hristache, et al. "Structure adaptive approach for dimension reduction." Annals of Statistics (2001): 1537-1566.
> 3. S. Trivedi,  et al. "A Consistent Estimator of the Expected Gradient Outerproduct." UAI. 2014.
> 4. Y. Xia, et al. "An adaptive estimation of dimension reduction space." Journal of the Royal Statistical Society Series B: Statistical Methodology 64.3 (2002): 363-410.
> 5. A. Radhakrishnan, et al. "Feature learning in neural networks and kernel machines that recursively learn features." arXiv preprint arXiv:2212.13881 (2022).
>
> *Q1: Any results on (stochastic) gradient descent with momentum?*
>
> In Section 4, we show that the correlation between the test performance and AGOP alignment holds for SGD+momentum and other optimization algorithms. This suggests that learning AGOP is useful to improve generalization with SGD+momentum.  As to the catapults in SGD with momentum, we believe it is an important direction of future work.
>
> *Q2: ...relations of AGOP to other "generalization measures" such as flatness (max eigenvalue of Hessian), tail-index of the trajectory/weight matrix [2,3]...etc*
>
> Connections to flat minima is an interesting and complicated issue.  There is evidence that flat minima lead to better generalization in, for example, multi-index models [1]. Yet, the same paper also provided examples for which flat minima do not always generalize.  Understanding connections between AGOP and flatness is an interesting area for future exploration.
>
> 1. W. Kaiyue, et al. "Sharpness minimization algorithms do not only minimize sharpness to achieve better generalization." arXiv preprint arXiv:2307.11007 (2023).
>
> *Q3: Any connection to recently uncovered saddle-to-saddle hopping dynamics [3,4]? Can the hoppings be equated to catapults?*
>
> In saddle-to-saddle hopping dynamics, there are no spikes in the training loss and these dynamics typically happen with small initialization. Thus, we believe there are no direct connections between these phenomena.

---

> > ### Comment · Reviewer_Ykc6 · 2023-11-23
> >
> > I've read through the author's response and the discussions with other reviewers, and I'm inclined to keep my score (leaning toward acceptance).
> >
> > Some additional minor comments:
> > - Figure 3(a) and Figure 3a have been mixed. It would be good for the formatting to be unified.
> > - in pg. 8, there is a "Figure ??"
> > - in pg. 3 edge of stability section, it should be "SGD (Kalra & Barkeshli, 2023)"

---

### Meta-Review · Area_Chair_bKq7 · 2023-12-21

**Metareview:**

After careful consideration of the reviewers' comments and the authors' responses, it is my decision to reject this submission. While the exploration of the catapult phenomenon in SGD and its relation to generalization is of interest to the community, the paper presents several fundamental weaknesses that have not been sufficiently addressed.

The primary concerns raised by the reviewers revolve around the lack of a clear theoretical framework and rigorous analysis to support the empirical observations. The paper falls short in providing a concrete explanation for why catapults occur and relies heavily on correlation rather than causation. Furthermore, the connection between the catapults and the top eigenspace of the NTK, while intriguing, is not fully expounded upon to demonstrate its significance or implications for deep learning.

Another significant concern is the paper's focus on the squared loss function, which does not align well with the common practice of using cross-entropy loss in classification tasks. This limits the paper's relevance to realistic deep learning settings and raises questions about the generalizability of the findings.

Additionally, there is a lack of clarity and coherence in the presentation of the experimental setup and results, which hampers the reproducibility and understandability of the work. The reviewers have also noted the absence of theoretical justification for the choice of step sizes and the claim that small batch SGD leads to better generalization, which is a contested point in the literature.

Despite the authors' efforts to address some of these concerns in their responses, the revisions have not been sufficient to overcome the fundamental issues. Therefore, the paper cannot be accepted in its current form.

**Justification For Why Not Higher Score:**

The decision to reject is based on the consensus of the reviewers that the paper has considerable weaknesses that are not adequately addressed by the authors. The lack of theoretical grounding, the focus on an unrealistic loss function setting, and issues with the clarity and presentation of the experiments are significant barriers to the paper's acceptance.

A higher score was not justified due to the paper's failure to provide a robust theoretical analysis or clear evidence of causation between catapults and generalization. The reviewers consistently pointed out these gaps, and the authors' responses did not satisfactorily resolve them.

**Justification For Why Not Lower Score:**

N/A

---

### Decision · Program_Chairs · 2024-01-16

Reject